# TMEM120A contains a specific coenzyme A-binding site and might not mediate poking- or stretch-induced channel activities in cells

Yao Rong[1,2†], Jinghui Jiang[3†], Yiwei Gao[1,2], Jianli Guo[1†], Danfeng Song[1,2], Wenhao Liu[3], Mingmin Zhang[3], Yan Zhao[1,2,4*], Bailong Xiao[3*], Zhenfeng Liu[1,2*]

[1]National Laboratory of Biomacromolecules, CAS Center for Excellence in Biomacromolecules, Institute of Biophysics, Chinese Academy of Sciences, Beijing, China; [2]College of Life Sciences, University of Chinese Academy of Sciences, Beijing, China; [3]State Key Laboratory of Membrane Biology; Tsinghua-Peking Center for Life Sciences; Beijing Advanced Innovation Center for Structural Biology; IDG/McGovern Institute for Brain Research; School of Pharmaceutical Sciences, Tsinghua University, Beijing, China; [4]State Key Laboratory of Brain and Cognitive Science, Institute of Biophysics, Chinese Academy of Sciences, Beijing, China

*For correspondence:
zhaoy@ibp.ac.cn (YZ);
xbailong@mail.tsinghua.edu.cn (BX);
liuzf@ibp.ac.cn (ZL)

†These authors contributed equally to this work

Competing interests: The authors declare that no competing interests exist.

**Abstract** TMEM120A, a member of the transmembrane protein 120 (TMEM120) family, has a pivotal function in adipocyte differentiation and metabolism, and may also contribute to sensing mechanical pain by functioning as an ion channel named TACAN. Here we report that expression of TMEM120A is not sufficient in mediating poking- or stretch-induced currents in cells and have solved cryo-electron microscopy (cryo-EM) structures of human TMEM120A (HsTMEM120A) in complex with an endogenous metabolic cofactor (coenzyme A, CoASH) and in the apo form. HsTMEM120A forms a symmetrical homodimer with each monomer containing an amino-terminal coiled-coil motif followed by a transmembrane domain with six membrane-spanning helices. Within the transmembrane domain, a CoASH molecule is hosted in a deep cavity and forms specific interactions with nearby amino acid residues. Mutation of a central tryptophan residue involved in binding CoASH dramatically reduced the binding affinity of HsTMEM120A with CoASH. HsTMEM120A exhibits distinct conformations at the states with or without CoASH bound. Our results suggest that TMEM120A may have alternative functional roles potentially involved in CoASH transport, sensing, or metabolism.

## Introduction

Through a fundamental cellular process known as mechanosensation, mechanical forces are converted into electrochemical signals during various physiological processes, such as touch, hearing, pain sensation, osmoregulation, and cell volume regulation (*Garcia-Anoveros and Corey, 1997*). In the past decades, various types of ion channels involved in sensing and transducing mechanical force signals have been identified (*Jin et al., 2020*). Among them, mechanosensitive channels of large and small conductances (MscL and MscS) serve as emergency release valves on the membrane by sensing and responding to membrane tension (*Booth and Blount, 2012*; *Haswell et al., 2011*). While members of the MscL family are present in bacteria, archaea, or fungi, and those of the MscS family exist in plants or microbes, they are not found in animals (*Booth et al., 2007*; *Pivetti et al., 2003*; *Wilson et al., 2013*). Piezo channels are involved in cellular mechanosensation processes crucial for touch perception, proprioception, red blood cell volume regulation, and other physiological

functions in animals and plants (*Mousavi et al., 2021*; *Murthy et al., 2017*; *Radin et al., 2021*; *Xiao, 2020*). Moreover, Drosophila NompC from the transient receptor potential (TRP) channel family, OSCA/TMEM63 (found in plants and animals) and TMC1/2 (mainly found in animals) from the TMEM16 superfamily, mammalian TREK/TRAAK channels from the two pore-domain potassium (K2P) channel subfamily, and *Caenorhabditis elegans* degenerin channels from the epithelial sodium channel (ENaC)/degenerin family also serve as mechanosensitive ion channels with pivotal functions in various mechanotransduction processes of eukaryotic organisms (*Jin et al., 2020*).

Transmembrane protein 120 (TMEM120) is a family of membrane proteins wide spread in animals and plants, and was originally identified as a nuclear envelope protein through a proteomic approach (*Malik et al., 2010*). Two paralogs of TMEM120 (TMEM120A and TMEM120B) exist in mammals, and they have crucial functional roles in adipocyte differentiation and metabolism (*Batrakou et al., 2015*). Knockout of *Tmem120a* in mice led to a lipodystrophy syndrome with insulin resistance and metabolic defects when the animals were exposed to a high-fat diet (*Czapiewski et al., 2021*). Recently, TMEM120A has been reported to function as an ion channel involved in sensing mechanical pain and thereby termed TACAN (*Beaulieu-Laroche et al., 2020*). The protein is expressed in a subset of nociceptors and is associated with mechanically evoked currents in heterologous cell lines. Besides, TACAN protein exhibited nonselective cation channel activity when heterologously expressed in cell lines or reconstituted in liposomes (*Beaulieu-Laroche et al., 2020*). Despite the previous report that suggested TACAN might function as a high-threshold mechanically activated cation channel responsible for sensing mechanical pain in mice, the mechanism of mechanosensation and ion permeation mediated by TACAN channel remains elusive and awaits further investigation. The molecular basis underlying the functions of TMEM120A in pain sensation and adipocyte differentiation is still unclear. Here we electrophysiologically characterized the ability of TMEM120A in mediating mechanically activated currents in cells and reconstituted proteoliposome membranes, and solved the cryo-electron microscopy (cryo-EM) structures of human TMEM120A (*Hs*TMEM120A) at two different conformational states. A striking discovery is revealed on the specific interactions between TMEM120A protein and an endogenous cofactor crucial for energy and fatty acid metabolism, namely CoASH (*Sibon and Strauss, 2016*), which provides important insights into the bona fide cellular functions of TMEM120 family.

## Results

### Functional characterizations of TMEM120A

When expressed heterologously in cell lines including CHO, HEK293T, and COS9, *Hs*TMEM120A only appeared to cause a pico-ampere level of increase in stretch-induced currents (*Beaulieu-Laroche et al., 2020*), prompting us to verify the role of TMEM120A in mediating mechanically activated (MA) currents. To exclude any endogenous Piezo1-mediated MA currents, we have chosen the previously reported *Piezo1* knockout HEK293T cells (P1-KO-HEK) (*Cahalan et al., 2015*) as our heterologous expression system. Using a piezo-driven blunted glass pipette to mechanically probe the cell membrane under a whole-cell recording configuration, we recorded robust MA currents from the P1-KO-HEK cells transfected with the construct of either Piezo1 or Piezo2 (*Figure 1A–C*). By contrast, similarly to vector-transfected cells, none of the cells transfected with the construct of either mouse TMEM120A (*Mm*TMEM120A, with or without C-terminal fusion of fluorescent protein; *Mm*TMEM120A-mCherry, with mCherry fused to the C-terminal region of *Mm*TMEM120A; *Mm*TMEM120A-ires-GFP, *Mm*TMEM120A and green fluorescent protein (GFP) were translated as two independent proteins not fused to each other, due to the presence of an internal ribosome entry site/ires) or *Hs*TMEM120A-mCherry showed poking-induced currents (*Figure 1A–C*). These data suggest that TMEM120A is not sufficient to mediate poking-induced currents. To examine whether TMEM120A might specifically mediate stretch-induced currents as originally reported (*Beaulieu-Laroche et al., 2020*), we measured stretch-induced currents by applying negative pressure from 0 to −120 mmHg to the membrane patch under either cell-attached (*Figure 1D–F*) or inside-out (*Figure 1G–I*) patch-clamp configurations. As a positive control, Piezo1 mediated robust stretch-induced currents under both cell-attached and inside-out recording configurations (*Figure 1D–I*). Consistent with the previous report that human TMEM63a (*Hs*TMEM63a) serves as a high-threshold mechanosensitive channel (*Murthy et al., 2018*), we indeed found that *Hs*TMEM63a-

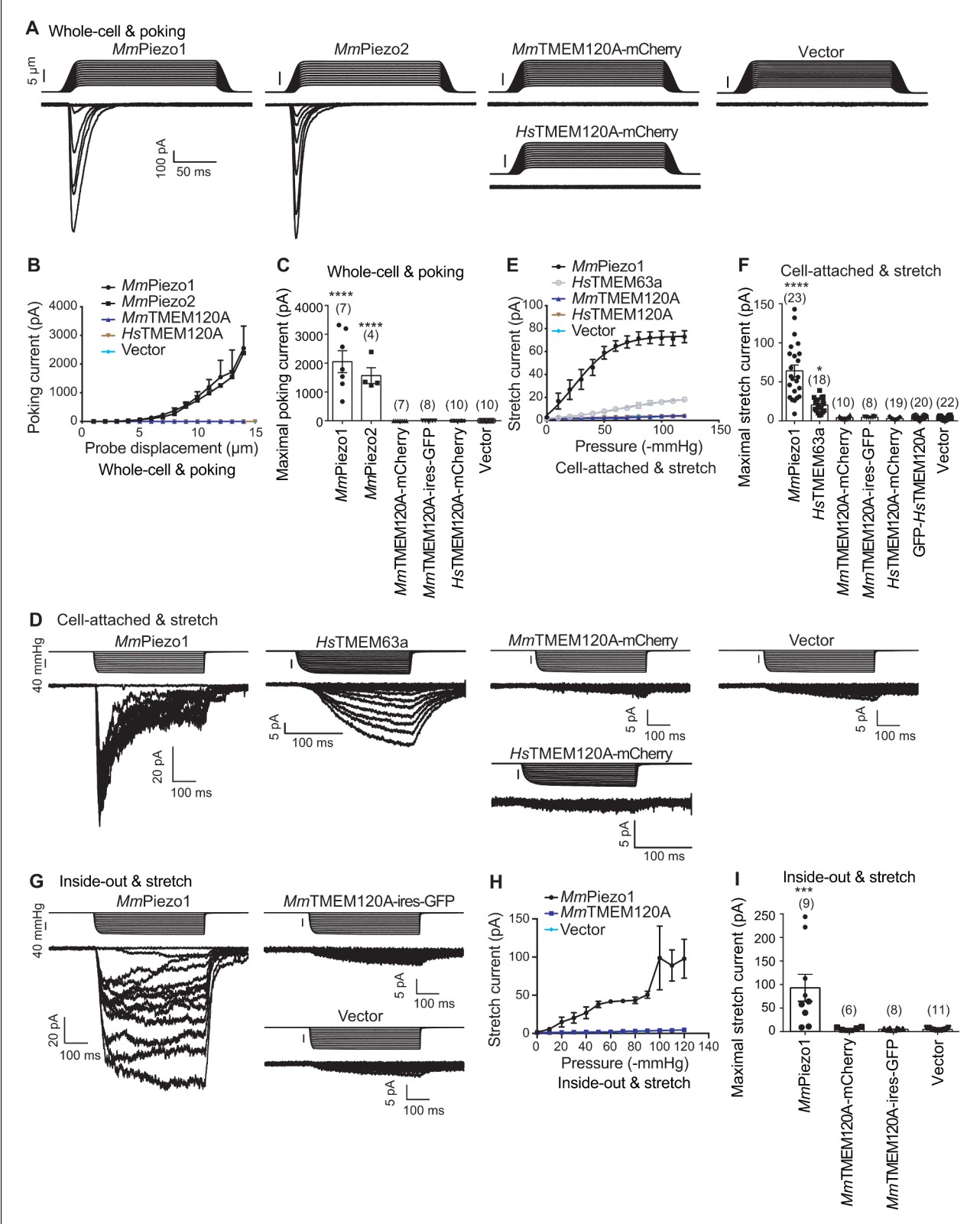

**Figure 1.** TMEM120A does not mediate poking- or stretch-induced currents in P1-KO-HEK cells. (**A**) Representative poking-evoked whole-cell currents from P1-KO-HEK cells transfected with the indicated constructs. (**B**) Current-displacement curves showing poking-evoked whole-cell currents from P1-KO-HEK cells transfected with the indicated constructs in response to the increased probe displacement steps. The *Mm*TMEM120A group includes data from both *Mm*TMEM120A-mCherry- and *Mm*TMEM120A-ires-GFP-transfected cells. (**C**) Scatter plot of the maximal poking-evoked whole-cell

*Figure 1 continued on next page*

*Figure 1 continued*

currents. (D) Representative stretch-induced currents from P1-KO-HEK cells transfected with the indicated constructs under the cell-attached patch configuration. (E) Current-pressure curves showing stretch-induced currents under cell-attached patch configuration from P1-KO-HEK cells transfected with the indicated constructs in response to the increased negative pressures. The *Mm*TMEM120A group includes data from both *Mm*TMEM120A-mCherry- and *Mm*TMEM120A-ires-GFP-transfected cells, while the *Hs*TMEM120A group includes data from both *Hs*TMEM120A-mCherry- and GFP-*Hs*TMEM120A-transfected cells. (F) Scatter plot of the maximal stretch-induced currents. (G) Representative stretch-induced currents from P1-KO-HEK cells transfected with the indicated constructs under the inside-out patch configuration. (H) Current-pressure curves showing stretch-induced currents under inside-out patch configuration from P1-KO-HEK cells transfected with the indicated constructs in response to the increased negative pressures. (I) Scatter plot of the maximal stretch-induced currents. In panels (C), (F), and (I), each bar represents mean ± sem, and the recorded cell number is labeled above the bar. One-way analysis of variance (ANOVA) with comparison to the vector. ***$p<0.001$; ****$p<0.0001$.

The online version of this article includes the following source data and figure supplement(s) for figure 1:

**Source data 1.** Source files for the electrophysiology data.

**Figure supplement 1.** Representative excised inside-out patch recordings of *Hs*TMEM120A reconstituted in GUVs.

**Figure supplement 1—source data 1.** Source file for the representative excised inside-out patch recordings shown in *Figure 1—figure supplement 1* A.

**Figure supplement 1—source data 2.** Source file for the representative excised inside-out patch recordings shown in *Figure 1—figure supplement 1* B.

**Figure supplement 1—source data 3.** Source file for the representative excised inside-out patch recordings shown in *Figure 1—figure supplement 1* C.

transfected cells reliably showed slowly activated stretch-induced currents, despite the maximal current amplitude being smaller than that of Piezo1-mediated currents (20.4 ± 2.2 vs 64.1 ± 7.4 pA) (*Figure 1D–F*). By contrast, under either cell-attached or inside-out patch configurations, none of the *Mm*TMEM120A- or *Hs*TMEM120A-transfected cells (a total of 57 cells) showed stretch-induced currents, similar to vector-transfected cells (*Figure 1D–I*). Given that none of the four different versions of TMEM120A constructs (*Mm*TMEM120A-mCherry, *Mm*TMEM120A-ires-GFP, *Hs*TMEM120A-mCherry, and GFP-*Hs*TMEM120A) exhibited poking- or stretch-induced currents in the cells, the lack of TMEM120A-mediated currents is highly unlikely due to the influence of the fusion tags. Thus, under our experimental conditions with the use of Piezo channels and *Hs*TMEM63a as proper positive controls, we conclude that TMEM120A is not sufficient to mediate poking- or stretch-induced currents in P1-KO-HEK cells.

Purified TMEM120A proteins were shown to mediate spontaneous channel activities with an estimated single-channel conductance of ~250 pS, which is drastically different from the single-channel conductance of 11.5 pS measured in cells (*Beaulieu-Laroche et al., 2020*). To verify whether purified TMEM120A proteins might mediate channel activities when reconstituted into lipid bilayers (*Beaulieu-Laroche et al., 2020*), we reconstituted purified *Hs*TMEM120A proteins into giant unilamellar vesicles (GUVs) and carried out single-channel recording on the inside-out excised membrane patch by applying a negative pressure on the membrane. As shown in *Figure 1—figure supplement 1*, out of over 200 (ie, 217) patches measured, only two exhibited pressure-dependent mini-conductance channel activities. In one patch, the channel became more active in response to the increasing negative pressure at both −80 and +80 mV (*Figure 1—figure supplement 1A*), whereas the other patch only responded to the increasing negative pressure at −80 mV but not at +80 mV (*Figure 1—figure supplement 1B*). Most other patches were either silent and insensitive to the negative pressure applied (*Figure 1—figure supplement 1C*) or exhibited leaky signals, suggesting that the overall channel activity of the protein reconstituted in GUVs is fairly low.

Taken together, our functional characterizations did not verify the previous report indicating that TMEM120A might form a mechanically activated ion channel.

## Overall structure of TMEM120A

To further investigate whether TMEM120A might form an ion channel or not, we went on to solve the cryo-EM structure of *Hs*TMEM120A with the hope to obtain hints about the function of TMEM120A through a structural approach. As shown in *Figure 2* (and *Figure 2—figure supplement 1*, *Supplementary file 1*), the cryo-EM structure of *Hs*TMEM120A protein reconstituted in lipid nanodiscs forms a homodimeric assembly with an overall shape resembling a seesaw rocker. The two monomers of *Hs*TMEM120A proteins are related by a twofold symmetry axis running through

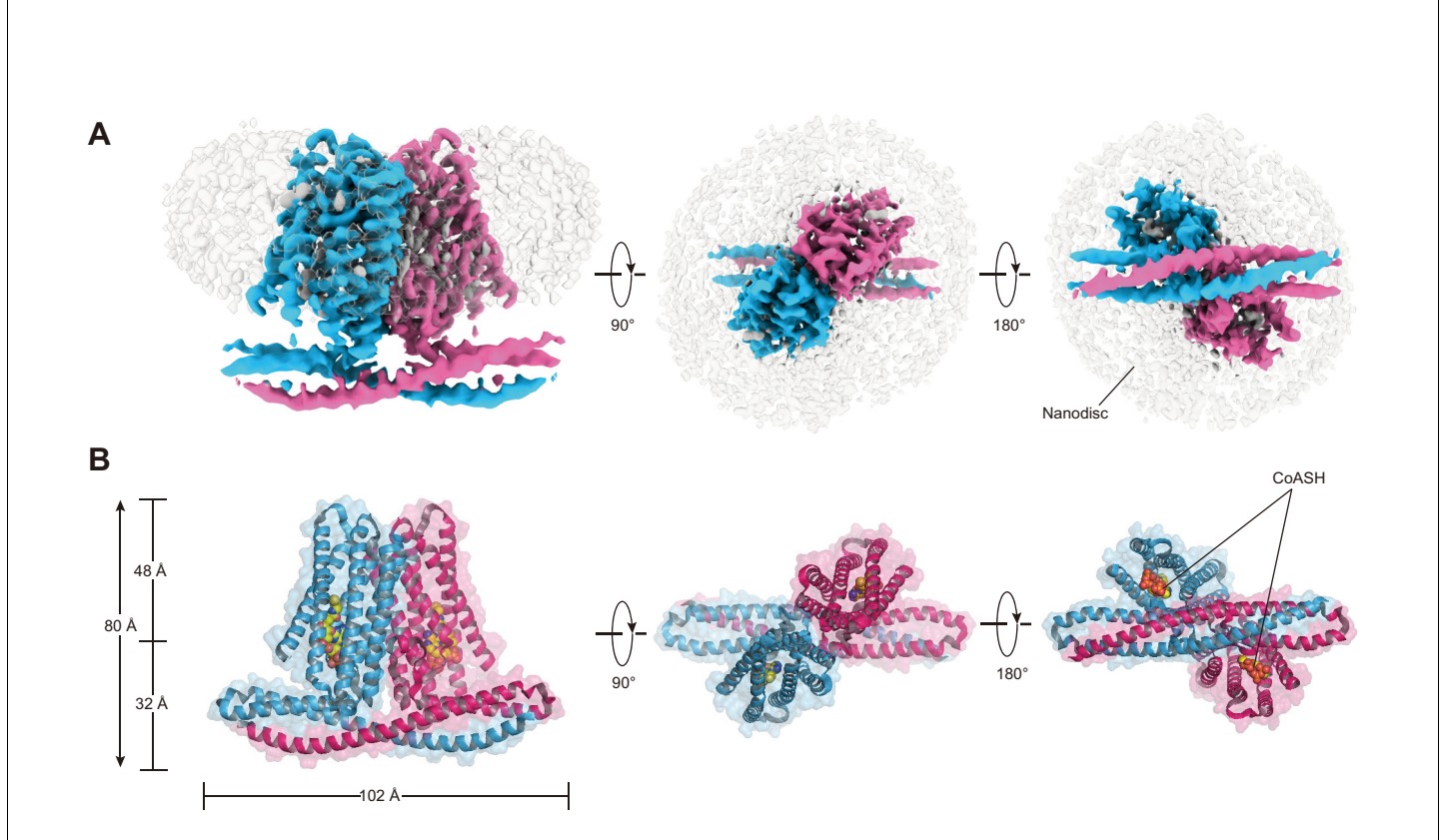

**Figure 2.** Cryo-EM density and overall architecture of *Hs*TMEM120A homodimer in complex with CoASH molecules. (**A**) Cryo-electron microscopy (cryo-EM) density of *Hs*TMEM120A-CoASH complex dimer embedded in a lipid nanodisc. The densities of two *Hs*TMEM120A protein subunits are colored blue and pink, while those of coenzyme A (CoASH) and the lipid nanodisc are colored silver. Side view along membrane plane, top view from extracellular side, and bottom view from intracellular side are shown from left to right. (**B**) Cartoon representations of the *Hs*TMEM120A-CoASH complex structure. The proteins are shown as cartoon models, whereas CoASH molecules are presented as sphere models. The views are the same as the corresponding ones in (**A**).

The online version of this article includes the following source data and figure supplement(s) for figure 2:

**Figure supplement 1.** Sample preparation, cryo-EM data collection, and processing of the *Hs*TMEM120A-CoASH complex reconstituted in nanodiscs.

**Figure supplement 1—source data 1.** Source file for the gel image data shown in Figure 2–figure supplement 1A.

their interface and perpendicular to the membrane plane. Each monomer contains an N-terminal soluble domain (NTD) on the cytosolic side and a transmembrane domain (TMD) embedded in a lipid bilayer (*Figure 3A*). There are two α-helices (intracellular helices 1 and 2, IH1 and IH2) in the NTD of each monomer (*Figure 3—figure supplement 1*). The long α-helix (IH1) of NTD intertwines with the symmetry-related one and interacts closely with the shorter one (IH2). IH1, IH2, and their symmetry-related ones in the adjacent monomer collectively form a coiled-coil structure with the long axis running approximately parallel to the membrane plane (*Figure 3B*). The TMD contains a bundle of six transmembrane helices (TM1-6) forming an asymmetric funnel with a wide opening on the intracellular side and a bottleneck on the extracellular side (*Figure 3D*). Between NTD and TMD, there is a hinge-like motif (HM) with two long loops and a short amphipathic α-helix positioned near the membrane surface on the intracellular side. The HM intercalates at the gap between the TMDs of two adjacent monomers and serves to stabilize the dimer by interacting with both subunits (*Figure 3C*).

## Identification of a CoASH-binding site in *Hs*TMEM120A

Unexpectedly, the *Hs*TMEM120A protein reconstituted in nanodiscs contains a CoASH molecule per monomer that is from an endogenous cellular source and copurified along with the protein (*Figure 4A*). Firstly, a small-molecule density is observed in the intracellular funnel-like cavity of each monomer and the density matches well with the CoASH model (*Figure 4B*) but not with the CoA

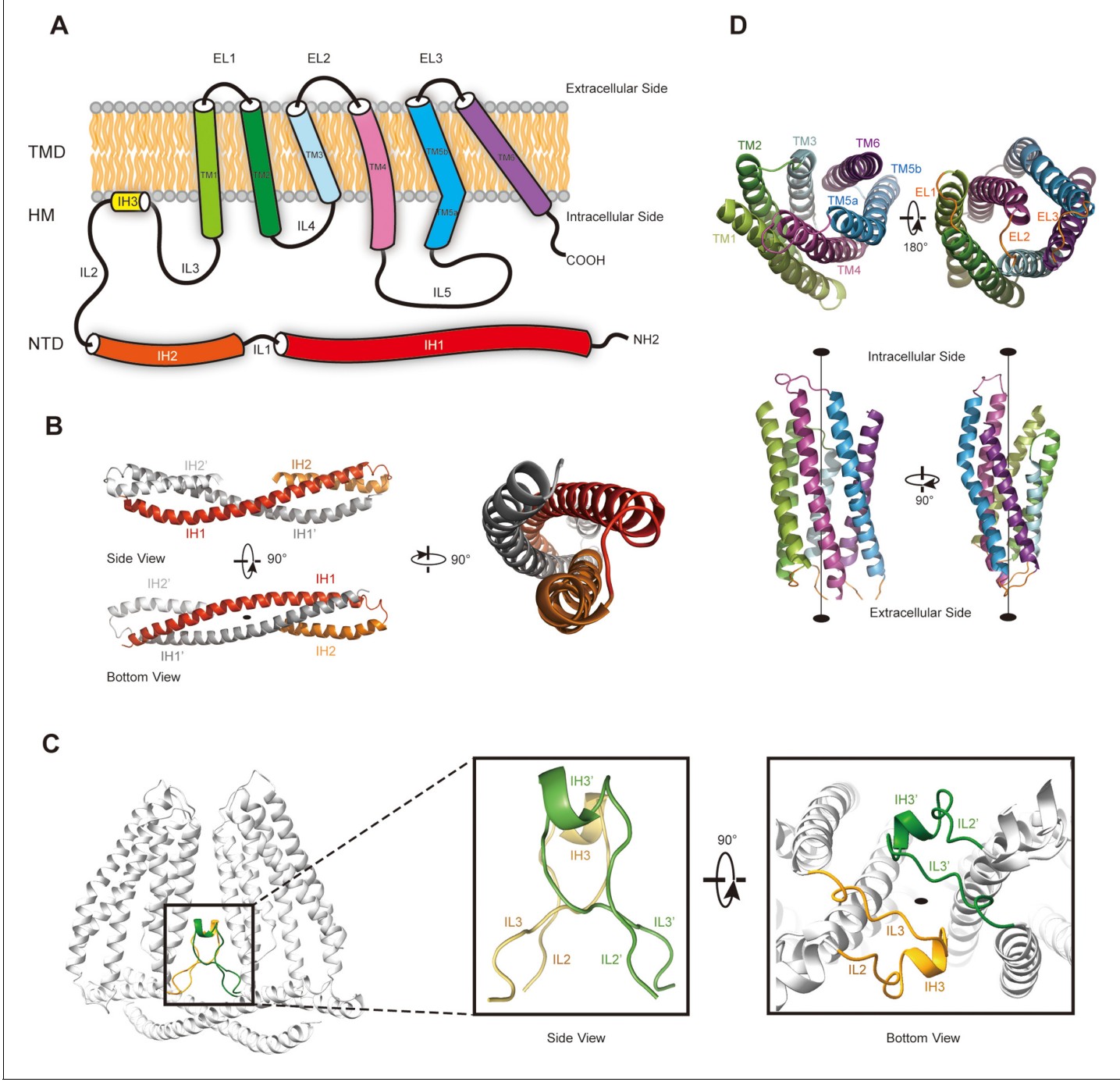

**Figure 3.** The membrane topology and domain structure of *Hs*TMEM120A monomer. (**A**) The topology of *Hs*TMEM120A monomer and arrangement of different parts relative to the membrane. The α-helices are presented as cylinder models. IH1-3, the intracellular helices 1–3; TM1-6, transmembrane helices 1–6; IL1-5, intracellular loops 1–6; EL1-3, extracellular loops 1–3. (**B**) The N-terminal domain (NTD) with two long and two short α-helices. (**C**) The role of hinge-like motif (HM) in mediating dimerization of *Hs*TMEM120A at the monomer-monomer interface. The solid elliptical rings in (**B**) and (**C**) indicate the central twofold axis of *Hs*TMEM120A homodimer. (**D**) The transmembrane domain with a bundle of six transmembrane helices. TM1, TM2, and TM3 are related to TM6, TM5, and TM4 through a pseudo-*C2* axis as indicated in the lower half.

The online version of this article includes the following figure supplement(s) for figure 3:

**Figure supplement 1.** Fitting of the structural model with the cryo-EM densities of various local regions of *Hs*TMEM120A in nanodiscs.

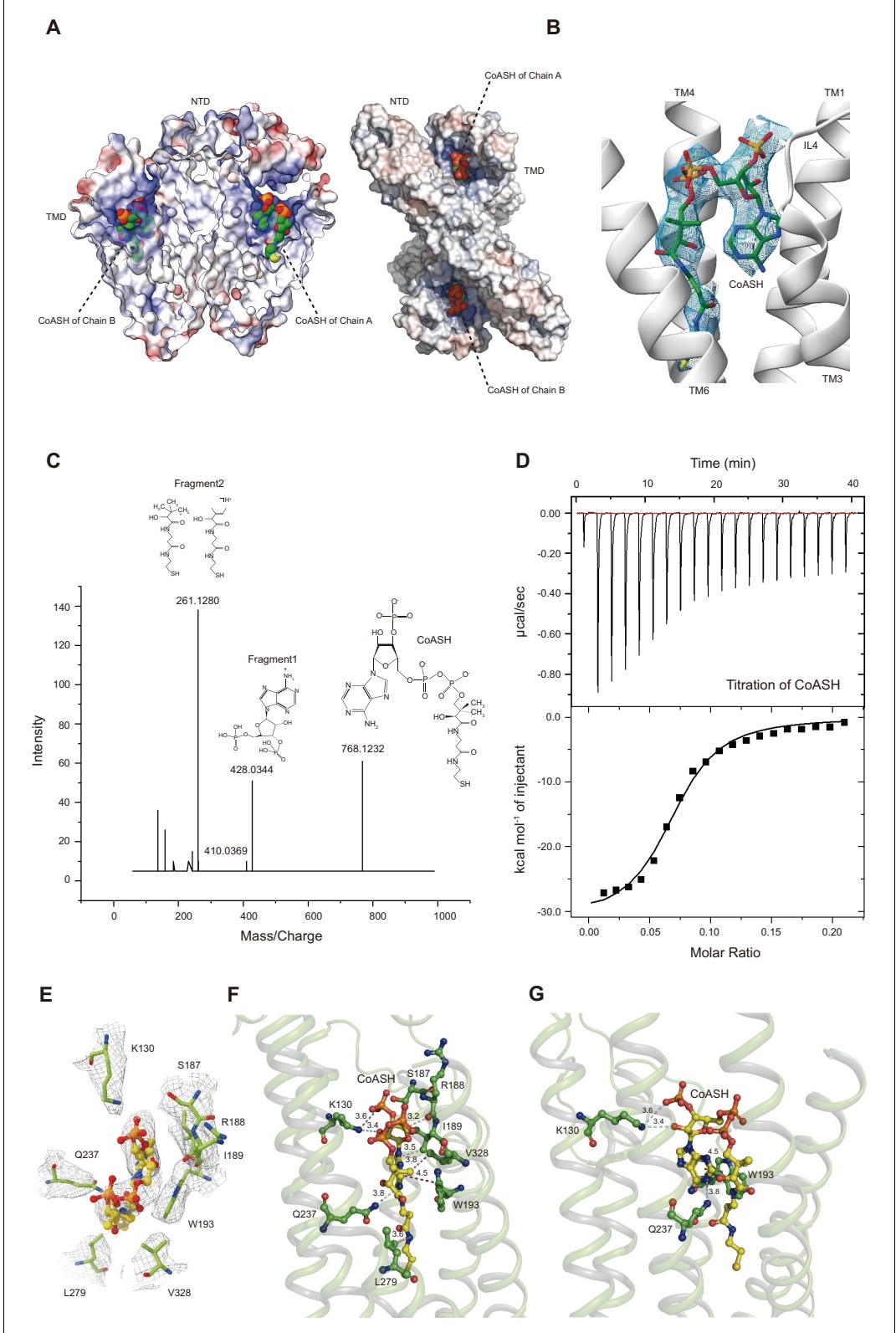

**Figure 4.** *Hs*TMEM120A contains an internal CoASH-binding site within each monomer. (**A**) Electrostatic potential surface presentation of *Hs*TMEM120A dimer reveals a deep coenzyme A (CoASH)-binding cavity with an electropositive surface. Left, side view; right, top view along the membrane normal from the intracellular side. The CoASH molecules are presented as sphere models. (**B**) The cryo-electron microscopy (cryo-EM) density of the ligand molecule bound to *Hs*TMEM120A fitted with a refined structural model of CoASH. (**C**) Mass spectrometry analysis of the small

*Figure 4 continued on next page*

*Figure 4 continued*

molecule extracted from the purified *Hs*TMEM120A protein. The chemical models of CoASH molecule and its fragments are shown above the corresponding peaks with m/z values 768.1232, 428.0344, and 261.1280. (D) Isothermal titration calorimetry analysis on the kinetic interactions between CoASH and *Hs*TMEM120A. The background heat of control was subtracted from the heat generated during binding of CoASH to *Hs*TMEM120A. The result is fit with the single-site-binding isotherm model with $\Delta H = -31.09 \pm 0.97$ kcal/mol and $K_d = 0.685 \pm 0.045$ μM. (E) Top view of the CoASH-binding site from the cytosolic side. The cryo-EM densities of CoASH and the surrounding amino acid residues (contoured at 9.5 rmsd) are superposed on the structural model. (F, G) Side views of the detailed interactions between CoASH and the adjacent amino acid residues of *Hs*TMEM120A from two different angles. The blue dotted lines indicate the hydrogen bonds or the salt bridge between Lys130 NZ and CoASH O2B (3.6 Å), Lys130 NZ and CoASH O8A (3.4 Å), and Gln237 NE2 and CoASH N1A (3.8 Å). The purple dotted line shows the π-π interaction between Trp193 and CoASH at 4.2–4.5 Å distances. The pink dotted lines exhibit the non-polar interactions between CoASH and the adjacent residues (Arg188, Ile189, Leu279, and Val328) at <4 Å distance.

The online version of this article includes the following source data and figure supplement(s) for figure 4:

**Figure supplement 1.** Fitting of the cofactor density with various small-molecule models.

**Figure supplement 2.** The conserved features of TMEM120A.

**Figure supplement 3.** Sequence alignment of *Hs*TMEM120A with various TMEM120B orthologs.

**Figure supplement 4.** Analysis of CoASH-binding property of the W193A mutant of *Hs*TMEM120A.

**Figure supplement 4—source data 1.** Source files for the ITC data, gel image, and western blot (*Figure 4—figure supplement 4A,B*).

derivatives (such as acetyl-CoA or fatty acyl CoA) or other nucleotide analogs (*Figure 4—figure supplement 1*). Secondly, mass spectrometry (MS) analysis on the perchloric acid extract of the purified *Hs*TMEM120A protein indicates that the protein does contain CoASH (*Figure 4C*). Thirdly, the CoASH molecule can bind to the ligand-free *Hs*TMEM120A protein with a dissociation constant ($K_d$) of 0.69 ± 0.05 μM (*Figure 4D*). The endogenous CoASH in the protein sample can be removed by purifying the protein through the size-exclusion chromatography in a detergent solution (see 'Materials and methods' for details), as indicated by the decrease of $A_{260}/A_{280}$ (absorbance at 260 nm/absorbance at 280 nm) value from 0.88 to 0.58 (note: while the protein absorption maximum is at ~280 nm, the absorption maximum of CoASH is at ~260 nm; *Zarzycki et al., 2008*). As shown in *Figure 4B*, the CoASH molecule assumes a bent conformation and forms numerous specific interactions with amino acid residues on the lumen surface of the cavity (*Figure 4E–G*). The amino acid residues involved in binding CoASH are highly conserved among various TMEM120A and TMEM120B homologs from different species (*Figure 4—figure supplements 2* and *3*). The 5-pyrophosphate and 3-phosphate groups of CoASH are positioned at the entrance of the cavity, while the adenosine and cysteamine groups of CoASH are inserted deep in the intracellular cavity pocket of *Hs*TMEM120A in a key-lock mode (*Figure 4B,E–G*). Trp193 of *Hs*TMEM120A forms a strong π-π stacking interaction with the adenine group of CoASH. Mutation of the Trp193 to Ala dramatically reduced the affinity between CoASH and the protein (*Figure 4—figure supplement 4*). Therefore, the TMD of *Hs*TMEM120A harbors a specific CoASH-binding site, potentially related to the function of TMEM120A protein.

## Conformational change of *Hs*TMEM120A upon dissociation of CoASH

Does *Hs*TMEM120A change its conformation upon dissociation of CoASH? To address this question, we solved the structure of *Hs*TMEM120A in a detergent micelle with no CoASH bound (*Figure 5* and *Figure 5—figure supplement 1A–D*, *Supplementary file 1*). The sample in the detergent exhibits an $A_{260}/A_{280}$ ratio much lower than the one in nanodiscs (*Figure 5—figure supplement 1E, F*), and no CoASH density is observed in the cavity (*Figure 5—figure supplement 1G,H*). In the CoASH-free structure (*Figure 5A and B*), the IL5 loop between TM4 and TM5 switches from the outward position to an inward position, partially covering the entrance of the intracellular cavity of *Hs*TMEM120A (*Figure 5C and D*). The absence of CoASH creates a spacious vacant lumen in the intracellular cavity of *Hs*TMEM120A protein, potentially allowing water and ions to enter through the cytoplasmic entrance (*Figure 5—figure supplement 2A*). Meanwhile, the cavity appears to be constricted at a narrow region on the extracellular side mainly by four amino acid residues, namely Met207, Trp210 and Phe219, and Cys310 (*Figure 5—figure supplement 2A–C*). Among them, only Trp210 is highly conserved across different species, while the other three are less conserved (*Figure 4—figure supplements 2* and *3*). In the presence of CoASH, the intracellular cavity of *Hs*TMEM120A is not only constricted by the four residues at the extracellular side but also blocked

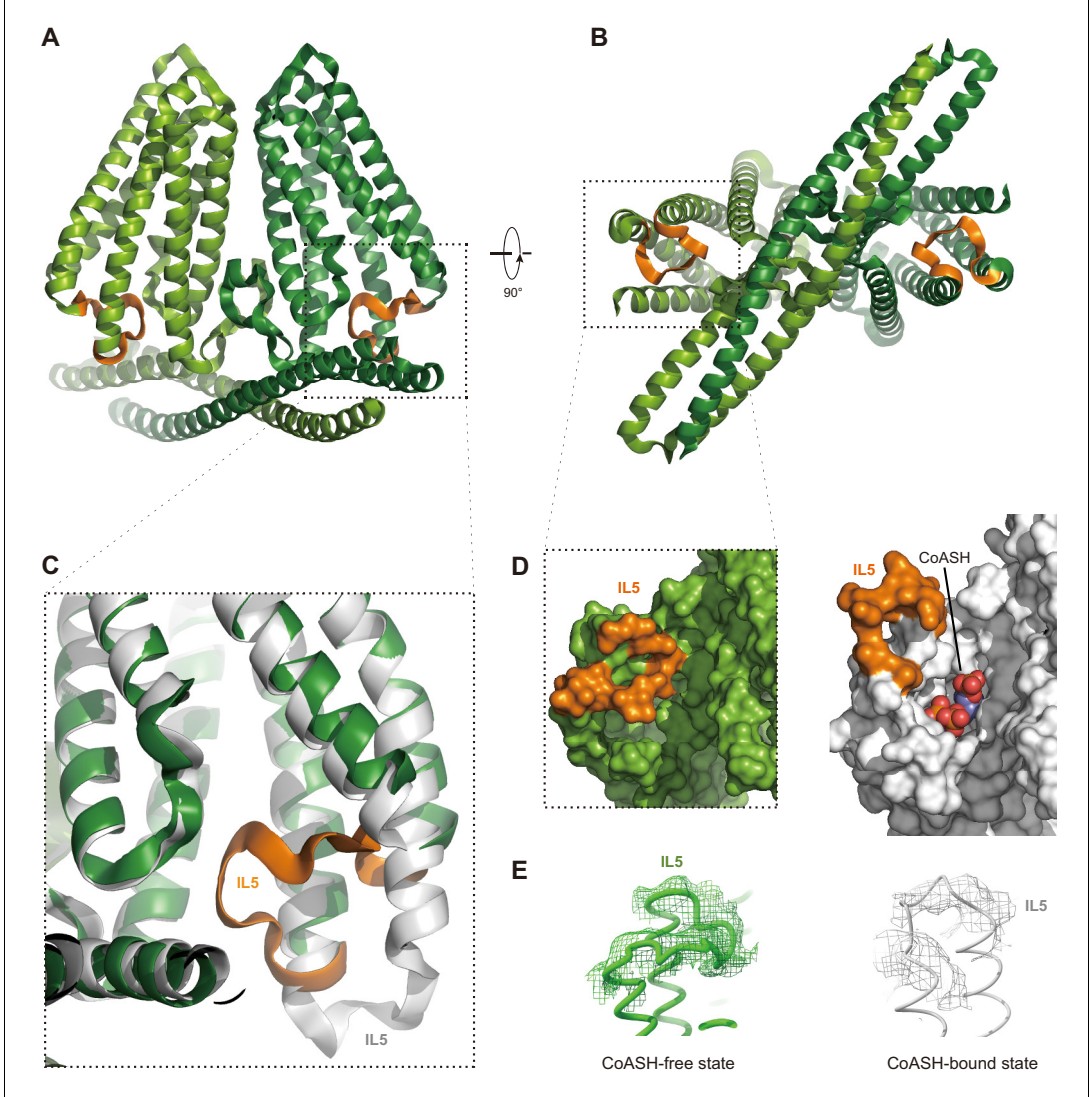

**Figure 5.** Structure of *Hs*TMEM120A at the CoASH-free state in comparison with the CoASH-bound state. (**A, B**) The overall structure of *Hs*TMEM120A without coenzyme A (CoASH) bound. The side view (**A**) and bottom view from intracellular side (**B**) are shown. The two monomers are colored light and dark green, respectively, while the intracellular loop IL5 is highlighted in orange. (**C**) Superposition of the structures of *Hs*TMEM120A at the CoASH-free state (green for the bulk region and orange for IL5) and CoASH-bound state (silver). The view is similar to the one in the dashed box of panel (**A**). (**D**) Surface presentation of the region around the CoASH-binding site in the CoASH-free (left) and CoASH-bound (right) *Hs*TMEM120A structures. The IL5 loop region is highlighted in orange. (**E**) Cryo-electron microscopy (cryo-EM) densities of the IL5 loop in the CoASH-free (left, contoured at 4.5 rmsd) and CoASH-bound (right, contoured at 3.3 rmsd) *Hs*TMEM120A structures.

The online version of this article includes the following figure supplement(s) for figure 5:

**Figure supplement 1.** Cryo-EM data collection and processing of *Hs*TMEM120A protein purified in detergent micelle.

**Figure supplement 2.** The occluded cavity of *Hs*TMEM120A with CoASH bound in comparison with the open cavity without CoASH bound.

by the bulky adenosine 3-phosphate 5-pyrophosphate group of CoASH as well as its cysteamine and pantothenate groups (*Figure 5—figure supplement 2A and B*).

## Discussion

In the previous report (*Beaulieu-Laroche et al., 2020*), heterologously expressed TMEM120A only mediated a picoampere-level of increase of stretch-induced currents on top of the endogenous currents and failed to respond to poking stimulation. However, it appeared to be more efficiently

activated via a pillar-based mechanical simulation at the cell-substrate interface. The single-channel conductance of the TMEM120A-mediated currents measured in cells was 11.5 pS. Surprisingly, reconstituted TMEM120A proteins were shown to mediate spontaneous channel activities with an estimated single-channel conductance of ~250 pS (*Beaulieu-Laroche et al., 2020*). The drastic difference in the single-channel conductance measured in cells and reconstituted lipid membranes raises the concern whether TMEM120A itself might indeed form a bona fide ion-conducting channel. Under our experimental conditions, when the purified *Hs*TMEM120A protein was reconstituted into GUVs, pressure-dependent channel activities were rarely observed during the single-channel recording processes. Moreover, we were unable to record TMEM120A-mediated poking- or stretch-induced currents in cells (*Figure 1*). We tested both *Mm*TMEM120A-mCherry and *Mm*TMEM120A-ires-GFP and observed neither poking- nor stretch-induced currents. Besides, we also tested both *Hs*TMEM120A-mCherry and GFP-*Hs*TMEM120A (*Hs*TMEM120A with GFP fused to its N-terminal region) and observed no stretch-induced currents. Thus, the lack of TMEM120A-mediated currents in our studies is not influenced by the presence or absence of a fusion tag (mCherry or GFP) at either the N-terminal or the C-terminal region. In line with the observation, both the N- and C-terminal regions of TMEM120A are suspended in cytosol and distant from the transmembrane domain, so that the fusion tags at the N- or C-terminal region might cause only minimal or no influence on protein function. Therefore, it is highly unlikely that the tag prevents the fusion protein from functioning as an ion channel as originally reported by *Beaulieu-Laroche et al., 2020*. Taken together, we conclude that TMEM120A is not sufficient to form a channel capable of sensing poking or stretching of the cell membrane. Nevertheless, we cannot completely rule out the possibility that TMEM120A might only respond to certain forms of mechanical stimuli such as perturbation at the cell-substrate interface, which might require further independent verification. In case that TMEM120A does function as a mechanosensitive channel, the CoASH molecule may serve as a plug to block the intracellular entry and consequently stabilize the channel in a closed state, while the intracellular cavity may become accessible to ions upon dissociation of CoASH molecule from the cavity (*Figure 5—figure supplement 2C*). When the channel is further activated by mechanical stimuli applied to the membrane, the constriction site on the extracellular side may open wider for ions to pass through the pore. However, the channel gating mechanism and coupling principle between putative CoASH inhibition and activation by mechanical stimuli during gating cycle would definitively need to be investigated further.

Recently, it has been reported that TMEM120 protein shares structural similarities with a membrane-embedded enzyme named elongation of very long chain fatty acid (ELOVL) protein (*Niu et al., 2021*; *Xue et al., 2021*). ELOVLs catalyze a condensation reaction between acyl-coA and malonyl-CoA to produce 3-keto acyl-CoA and release CoASH as well as $CO_2$ as byproducts (*Nie et al., 2021*). Although TMEM120A did not exhibit ELOVL-like enzymatic activity when malonyl-CoA and stearoyl-CoA were supplied as substrates (*Niu et al., 2021*), it is possible that TMEM120A may utilize different substrates or catalyze a reaction distinct from that of ELOVLs if it functions as an enzyme. In *C. elegans*, the function of TMEM120 is related to incorporation of fatty acids into triacylglycerol, and deficiency of TMEM120 in *C. elegans* is associated with a reduction of triacylglycerol level and lipid droplet size (*Li et al., 2021*). Co-expression of TMEM120A with Piezo2 reduced mechanically activated currents, and it was suggested that TMEM120A may function as a negative modulator of Piezo2 channel activity by modifying the lipid content of the cell (*Del Rosario et al., 2021*). Therefore, other potential functions of TMEM120A, such as being a membrane-embedded enzyme utilizing CoASH as a substrate or a cofactor, a CoASH transporter or a CoASH-sensing receptor, await to be explored further in the future.

## Materials and methods

### Key resources table

| Reagent type (species) or resource | Designation | Source or reference | Identifiers | Additional information |
|---|---|---|---|---|

*Continued on next page*

*Continued*

| Reagent type (species) or resource | Designation | Source or reference | Identifiers | Additional information |
|---|---|---|---|---|
| Gene (*Homo sapiens*) | *Hs*TMEM120A-pFastBac Dual | Synthetic/ Genscript | N/A | Custom-synthesized cDNA |
| Gene (*H. sapiens*) | *Hs*TMEM120A-mCherry-pDEST | Yulong Li lab in Peking University | N/A | |
| Gene (*Mus musculus*) | *Mm*TMEM120A-pEG BacMam | Synthetic | N/A | |
| Gene (*M. musculus*) | *Mm*TMEM120A-pcDNA3.1 | Synthetic | N/A | FP: gccctctagactc gagcggccgcgccacc ATGCAGTCCCCG CCCCCGGAC RP: GGCGCGCCAAGCT TCTAGTCCTTCTTGT TCCCGTGCTGCTGGCTG |
| Gene (*M. musculus*) | *Mm*Piezo1-mRuby2 pcDNA3.1 | *Zhao et al., 2018* | N/A | |
| Gene (*M. musculus*) | *Mm*Piezo2-GST-ires-GFP pcDNA3.1 | *Wang et al., 2019* | N/A | |
| Gene (*H. sapiens*) | *Hs*TMEM63A-mCherry-pDEST | Yulong Li lab in Peking University | N/A | |
| Strain, strain background (*Escherichia coli*) | Turbo Chemically Competent Cell | AngYuBio | Cat# X17012 | |
| Strain, strain background (*E. coli*) | DH10Bac Chemically Competent Cell | AngYuBio | Cat# G6006 | |
| Cell line (*Spodoptera frugiperda*) | Sf9 | ATCC | Cat# CRL-1711 RRID:CVCL_0549 | |
| Cell line (*H. sapiens*) | HEK293T-*Piezo1*-KO | *Cahalan et al., 2015* | N/A | |
| Antibody | THE NWSHPQFEK Tag Antibody [HRP] (mouse monoclonal) | Genscript | Cat# A01742 RRID:AB_2622218 | WB (1:10,000) |
| Recombinant DNA reagent | *Hs*TMEM120A W193A-pFastBac Dual | This study | N/A | Construct made and maintained in Z Liu lab |
| Sequence-based reagent | W193A _Forward | Beijing Genomics Institution | PCR primers | CGTATCAAGGGGTTG GGCCGTGTTC CACCACTAC |
| Sequence-based reagent | W193A_Reverse | Beijing Genomics Institution | PCR primers | GTAGTGGTGG AACACG GCCCAACCCT TGATACG |
| Chemical compound, drug | Cellfectin II Reagent | Gibco | Cat# 11605102 | |
| Chemical compound, drug | ESF 921 Insect Cell Culture Medium, Protein Free | Expression Systems | Cat# 96-001-01 | |
| Chemical compound, drug | Grace's Insect Medium, supplemented | Gibco | Cat# 11605102 | |
| Chemical compound, drug | n-Dodecyl-β-D-Maltopyranoside (β-DDM) | Anatrace | Cat# D310 | |

*Continued on next page*

Continued

| Reagent type (species) or resource | Designation | Source or reference | Identifiers | Additional information |
|---|---|---|---|---|
| Chemical compound, drug | Cholesteryl Hemisuccinate (CHS) | Anatrace | Cat# CH210 | |
| Chemical compound, drug | CHAPS | Anatrace | Cat# C316 | |
| Chemical compound, drug | GDN | Anatrace | Cat# GDN101 | |
| Chemical compound, drug | Streptavidin Beads 6FF | Smart Lifesciences | Cat# SA021100 | |
| Chemical compound, drug | d-Desthiobiotin | Sigma-Aldrich | Cat# D1411 | |
| Chemical compound, drug | 18:1 (Δ9-Cis) PE (DOPE) | Avanti | Cat# 850725 | Powder |
| Chemical compound, drug | 16:0-18:1 PS (POPS) | Avanti | Cat# 840034 | Powder |
| Chemical compound, drug | 16:0-18:1 PC (POPC) | Avanti | Cat# 850457 | Powder |
| Chemical compound, drug | Bio-Beads SM-2 adsorbents | Bio-Rad | Cat# 1523920 | |
| Chemical compound, drug | Perchloric acid | SCR | Cat# 10015160 | |
| Chemical compound, drug | Coenzyme A sodium salt hydrate | Sigma-Aldrich | Cat# C4780 | |
| Chemical compound, drug | Lipofectamine 3000 transfection kit | ThermoFisher | Cat# L3000015 | |
| Chemical compound, drug | poly-D-lysine | Beyotime | Cat# C0312 | |
| Chemical compound, drug | L-α-phosphatidyl choline | Sigma-Aldrich | Cat# P3644 | Powder |
| Chemical compound, drug | Cholesterol | Sigma-Aldrich | Cat# C8667 | Powder |
| Chemical compound, drug | 4ME 16:0 PC (DPhPC) | Avanti | Cat# 850356 | Powder |
| Software, algorithm | Serial EM | http://bio3d.colorado.edu/SerialEM | RRID:SCR_017293 | https://doi.org/10.1016/j.jsb.2005.07.007 |
| Software, algorithm | MotionCor2 | http://msg.ucsf.edu/em/software/motioncor2.html | RRID:SCR_016499 | https://doi.org/10.1038/nmeth.4193 |

*Continued*

| Reagent type (species) or resource | Designation | Source or reference | Identifiers | Additional information |
|---|---|---|---|---|
| Software, algorithm | CTFFIND 4.1.10 | http://grigoriefflab.janelia.org/ctffind4 | RRID:SCR_016732 | https://doi.org/10.1016/j.jsb.2015.08.008 |
| Software, algorithm | cryoSPARC v3.1 | https://cryosparc.com/ | RRID:SCR_016501 | https://doi.org/10.1038/nmeth.4169 |
| Software, algorithm | Chimera | https://www.cgl.ucsf.edu/chimera/download.html | RRID:SCR_004097 | https://doi.org/10.1002/jcc.20084 |
| Software, algorithm | Gctf | https://www.mrclmb.cam.ac.uk/kzhang/Gctf/ | RRID:SCR_016500 | https://doi.org/10.1016/j.jsb.2015.11.003 |
| Software, algorithm | Gautomatch | https://hpc.nih.gov/apps/gautomatch.html | N/A | http://www.mrc-lmb.cam.ac.uk/kzhang/ |
| Software, algorithm | Topaz | http://cb.csail.mit.edu/cb/topaz/ | N/A | https://doi.org/10.1038/s41592-019-0575-8 |
| Software, algorithm | cisTEM | https://cistem.org/ | RRID:SCR_016502 | https://doi.org/10.7554/eLife.35383 |
| Software, algorithm | MonoRes | http://scipion.i2pc.es/ | N/A | https://doi.org/10.1016/j.str.2017.12.018 |
| Software, algorithm | COOT 0.8.9 | http://www2.mrclmb.cam.ac.uk/personal/pemsley/coot | RRID:SCR_014222 | https://doi.org/10.1107/S0907444910007493 |
| Software, algorithm | PSIPRED 4.0 | http://bioinf.cs.ucl.ac.uk/psipred | RRID:SCR_010246 | https://doi.org/10.1093/nar/gkz297 |
| Software, algorithm | PHENIX | https://www.phenix-online.org | RRID:SCR_014224 | https://doi.org/10.1107/S0907444909052925 |
| Software, algorithm | HOLE | http://www.holeprogram.org/ | N/A | https://doi.org/10.1016/s0263-7855(97)00009-x |
| Software, algorithm | ChimeraX | https://www.rbvi.ucsf.edu/chimerax/ | RRID:SCR_015872 | https://doi.org/10.1002/pro.3943 |
| Software, algorithm | PyMol 2.1.1 | Schrödinger, LLC | RRID:SCR_000305 | https://pymol.org/ |
| Software, algorithm | Peakview 2.1 | https://sciex.com/products/software/peakview-software | RRID:SCR_015786 | |
| Software, algorithm | MicroCal ITC200 | https://www.malvernpanalytical.com/ | RRID:SCR_020260 | |
| Software, algorithm | ESPript 3 | https://espript.ibcp.fr | RRID:SCR_006587 | https://doi.org/10.1093/nar/gku316 |
| Software, algorithm | T-COFFEE | http://tcoffee.crg.cat/apps/tcoffee/index.html | RRID:SCR_019024 | https://doi.org/10.1006/jmbi.2000.4042 |
| Software, algorithm | The ConSurf Server | https://consurf.tau.ac.il/ | RRID:SCR_002320 | https://doi.org/10.1093/nar/gkw408 |
| Software, algorithm | VMD 1.9.3 | https://www.ks.uiuc.edu/Research/vmd/ | RRID:SCR_001820 | |

*Continued on next page*

*Continued*

| Reagent type (species) or resource | Designation | Source or reference | Identifiers | Additional information |
|---|---|---|---|---|
| Software, algorithm | PatchMaster | HEKA http://www.heka.com/downloads/downloads_main.html#down_patchmaster | RRID:SCR_000034 | |
| Software, algorithm | GraphPad Prism | GraphPad Software Inc http://www.graphpad.com/scientific-software/prism/ | RRID:SCR_002798 | |
| Software, algorithm | Origin 9.2 | OriginLab http://www.originlab.com/ | RRID:SCR_014212 | |
| Software, algorithm | Igor Pro | WaveMetrics https://www.wavemetrics.com/ | RRID:SCR_000325 | |
| Other | Structural model of human TMEM120A in the CoASH-bound state and the corresponding cryo-EM map | This study | PDB ID 7F3T, EMD-31440 | Available for download in the webpage https://www.rcsb.org/structure/7F3T |
| Other | Structural model of human TMEM120A in the CoASH-free state and the corresponding cryo-EM map | This study | PDB ID 7F3U, EMD-31441 | Available for download in the webpage https://www.rcsb.org/structure/7F3U |
| Other | R1.2/1.3 100 Holey Carbon Films Cu 200 mesh | Quantifoil | Cat# Q56244 | |
| Other | R1.2/1.3 100 Holey Carbon Films Cu 300 mesh | Quantifoil | Cat# Q55987 | |
| Other | Immobilon-P$^{SQ}$ transfer membrane | Merck Millipore | Cat# ISEQ00010 | Filter type: PVDF |

## Constructs and molecular cloning

The *Hs*TMEM120A-mCherry and *Hs*TMEM63A-mCherry constructs used for electrophysiological experiments were cloned from the complementary DNA (cDNA) library of human ORF 8.1 (a gift from Yulong Li's lab at the Peking University). Through gateway reactions, the coding sequence of *Hs*TMEM120A or *Hs*TMEM63A was cloned into the pDEST-mCherry expression vector, resulting in fusion of mCherry at the C-terminus of *Hs*TMEM120A or *Hs*TMEM63A. Compared to the amino acid sequence of *Hs*TMEM120A reported by *Beaulieu-Laroche et al., 2020*, the *Hs*TMEM120A-mCherry construct contained a natural variation of residue 201 (A201 instead of T201). To exclude the possibility that this variation might affect mechanically activated channel activities of *Hs*TMEM120A, we also synthesized the cDNA of *Hs*TMEM120A (with codons optimized for expression in HEK293 cells) encoding the exact same amino acid sequence as that reported by *Beaulieu-Laroche et al., 2020* and cloned it into the pcDNA3.1 vector with an upstream GFP-encoding sequence (GFP-*Hs*TMEM120A). Both *Hs*TMEM120A (T201A variant)-mCherry and GFP-*Hs*TMEM120A were tested for the function in mediating stretch-induced currents. The cDNA of *Mm*TMEM120A was amplified from a mouse cDNA library and subcloned into modified pEG BacMam vectors, yielding two constructs with the coding regions of mCherry-StrepII and ires-GFP being fused to the 3'-region of the target gene, respectively (*Mm*TMEM120A-mCherry and *Mm*TMEM120A-ires-GFP). The constructs

were verified by sequencing. *Mm*Piezo1-mRuby2 and *Mm*Piezo2-GST-ires-GFP were reported in our previous studies (*Geng et al., 2020*; *Wang et al., 2019*; *Zhao et al., 2018*).

To express *Hs*TMEM120A protein in the *Spodoptera frugiperda* Sf9 insect cells (RRID:CVCL_0549) for cryo-EM, isothermal titration calorimetry (ITC), and single-channel electrophysiology studies, the cDNA encoding the full-length *Homo sapiens* TMEM120A (*Hs*TMEM120A) was synthesized with codon optimization for *S. frugiperda* (Genescript). The cDNA was cloned into pFastBac Dual vector between the 5' BamH1 and 3' EcoR1 sites, and an N-terminal FLAG followed by twin-Strep-tags was fused with the target protein in the product.

## Protein expression and purification

After purification, the *Hs*TMEM120A-pFastBac Dual plasmid DNA was used to transform DH10Bac cells to generate the recombinant bacmid. Positive clones containing desired genes were identified by blue/white selection, and the recombinant bacmids were amplified and purified under sterile conditions. After three generations of amplification, the recombinant baculoviruses were used for transfection of Sf9 cells adapted to suspension culture. To increase protein expression level, the Sf9 cells were cultured at 27℃ for 24 hr and then at 20℃ for 48 hr before being harvested.

The cells were harvested from the cell culture medium through centrifugation at 2500 $\times g$ and ~10 g cell pellets were obtained from 800 ml cell culture. The cell pellets were frozen and stored at ~80℃ for later use. After having defrosted, the cell pellets were resuspended in 135 ml lysis buffer with 400 mM NaCl, 50 mM 4-(2-hydroxyethyl)-1-piperazineethanesulfonic acid (HEPES) (pH 7.5), 10% glycerol, and Protease Inhibitor Cocktail (MedChemExpress, cat. no.: HY-K0010; 1 ml 100× stock solution per 150 ml solution), and then disrupted by a high-pressure homogenizer (ATS Engineering Inc; Nano Homogenize Machine) under 1200 bar for six to eight cycles. Unless stated otherwise, all steps of protein purification were conducted at 4℃. After the cells were lysed, 15 ml of the detergent mixture stock solution with 10% n-dodecyl-β-D-maltopyranoside (β-DDM) and 2% cholesteryl hemisuccinate (CHS) was added into the lysate. The full-length *Hs*TMEM120A was extracted from the membrane of 10–15 g Sf9 cells (harvested from 800 ml culture) for 1 hr in 150 ml of the solubilization solution with 360 mM NaCl, 45 mM HEPES (pH 7.5), 1 mM ethylenediaminetetraacetic acid (EDTA) (pH 7.5), Protease Inhibitor Cocktail, 1% β-DDM, and 0.2% CHS. In order to remove the insoluble fraction, the sample was centrifuged at 39,191 $\times g$ (JL-25.50 rotor; Beckman) for 30 min. The supernatant was incubated with 5 ml Streptactin beads for 2 hr and then the beads were loaded into a 20 ml chromatography column. The flow through was discarded and five column volumes (CV) of washing buffer with 50 mM NaCl, 20 mM HEPES (pH 7.5), 1 mM EDTA (pH 7.5), Protease Inhibitor Cocktail (MedChemExpress; 1 ml 100× stock solution per 150 ml solution), 0.006% glyco-diosgenin (GDN), 0.006% 3-cholamidopropyl dimethylammonio 1-propanesulfonate (CHAPS), and 0.001% CHS was applied to the column to remove contaminant proteins. Subsequently, the target recombinant protein was eluted in the elution buffer (5 CV) containing 50 mM NaCl, 20 mM HEPES (pH 7.5), 1 mM EDTA (pH 7.5), Protease Inhibitor Cocktail (MedChemExpress; 1 ml 100× stock solution per 150 ml solution), 0.006% GDN, 0.006% CHAPS, 0.001% CHS, and 2.5 mM d-desthiobiotin. The purified protein sample was concentrated in a 100-kDa molecular weight cut-off (MWCO) concentrator (Millipore) to 3 mg/ml by centrifugation at 2500 $\times g$. Each batch of sf9 cells from 800 ml culture usually yielded 3–6 mg *Hs*TMEM120A protein through the above purification process.

The presence of CoASH in the purified *Hs*TMEM120A sample was monitored by measuring the absorbance under 260 nm and 280 nm ($A_{260}$ and $A_{280}$) of the protein sample with Nanodrop 2000 (Thermo). As the absorption peak of CoASH (copurified with the protein) is at ~260 nm (*Zarzycki et al., 2008*), a relatively high $A_{260}/A_{280}$ value of 0.88–0.96 was measured with the purified *Hs*TMEM120A samples from different preparations. To further improve the homogeneity of the purified protein, the concentrated protein sample was loaded onto a Superdex 200 increase 10/300 GL size-exclusion column (GE Healthcare Life Sciences) and eluted in the gel-filtration (GF) buffer containing 50 mM NaCl, 20 mM HEPES (pH 7.5), 1 mM EDTA (pH 7.5), Protease Inhibitor Cocktail, 0.006% GDN, 0.006% CHAPS, and 0.001% CHS. The peak fraction was collected and concentrated to 2.5–3.5 mg ml$^{-1}$ in a 100-kDa MWCO Amicon concentrator (Millipore). The $A_{260}/A_{280}$ value of the protein was lowered to 0.58–0.62 after size-exclusion chromatography.

For reconstitution of the *Hs*TMEM120A protein in nanodiscs, the protocol for protein solubilization and binding to the Streptactin resin as well as column washing was the same as the one

described above. During the washing step, 5 ml washing buffer was added into the column so that the final volume of the mixture was made ~10 ml (containing 5 ml Streptactin beads). For nanodisc preparation, membrane scaffold protein 1E3D1 (MSP1E3D1) was expressed in *Escherichia coli* cell, purified through immobilized metal affinity chromatography, and the His-tag was cleaved by the Tobacco Etch Virus (TEV) protease. The optimal ratio of *Hs*TMEM120A monomer:MSP1E3D1:phospholipid was 1:2:150 (molar ratio). Initially, 2 ml of 10 mM phospholipid mixture (1,2-dioleoyl-sn-glycero-3-phosphoethanolamine/DOPE:1-palmitoyl-2-oleoyl-sn glycero-3-phospho-L-serine/POPS:1-palmitoyl-2-oleoyl-glycero-3-phosphocholine/POPC = 2:1:1, molar ratio, solubilized in 1% GDN and 2% β-DDM with ddH$_2$O) was added into the column loaded with *Hs*TMEM120A protein and incubated for 30 min with constant rotation. Subsequently, the scaffold protein (MSP1E3D1) and bio-beads (100 mg) were added into the column to trigger formation of nanodiscs by removing detergents from the system. The mixture was incubated for 2 hr with constant rotation. Another batch of bio-beads (50 mg) was subsequently added and the mixture was incubate for one more hour. To remove the residual components and empty the nanodiscs (without target protein), the column was mounted vertically on the stand and the solution on the column was separated from the resin through gravity flow. Afterwards, the resin on the column was washed with 10 CV of washing buffer with 50 mM NaCl, 20 mM HEPES (pH 7.5), 1 mM EDTA (pH 7.5), and Protease Inhibitor Cocktail. The *Hs*TMEM120A-nanodisc complex assembled on the Streptactin beads was eluted by applying 5 CV of elution buffer with 50 mM NaCl, 20 mM HEPES (pH 7.5), 1 mM EDTA (pH 7.5), and Protease Inhibitor Cocktail. The purified nanodisc sample was concentrated with a 100-kDa MWCO concentrator to 3.5 mg/ml through centrifugation at 2500 $\times g$. To analyze the homogeneity of nanodiscs, the sample was run through size-exclusion chromatography (Superdex 200 increase 10/300 GL) and eluted in a solution with 50 mM NaCl, 20 mM HEPES (pH 7.5), 1 mM EDTA (pH 7.5), and Protease Inhibitor Cocktail. The major peak fraction was collected and concentrated to 3.5–4.5 mg ml$^{-1}$ in a 100-kDa MWCO Amicon concentrator. The purity of the protein sample was assessed by sodium dodecyl sulfate-polyacrylamide gel electrophoresis (SDS-PAGE). For western blots, the NWSHPQFEK tag antibody (HRP) (Genscript, 1:10,000 dilution; RRID:AB_2622218) was used to probe the target protein bands on the polyvinylidene fluoride (PVDF) membranes (Merck-Millipore) after the proteins were transferred from SDS-PAGE gels to the membranes.

## Cryo-EM sample preparation and data collection

For cryo-EM sample preparation, 3 µl of purified *Hs*TMEM120A sample was applied to the grid glow-discharged in H$_2$/O$_2$ for 60 s (Gatan Solarus 950). The Quantifoil 1.2/1.3 µm holey carbon grids (300 mesh, copper) were selected for preparing the *Hs*TMEM120A sample in the detergent, and 200 mesh grids of the same type were used for *Hs*TMEM120A-nanodisc sample. After having applied to the grids, the samples were blotted for 8 s with a Vitrobot (Vitrobot Mark IV; Thermo Fisher) after waiting for 10 s in the chamber at 4℃ with 100% humidity. The grid was plunge frozen in liquid ethane pre-cooled by liquid nitrogen. Cryo-EM images of the *Hs*TMEM120A particles in the detergent were collected on a Titan Krios transmission electron microscope operated at 300 kV. The images were recorded by using SerialEM program (*Mastronarde, 2003*) (RRID:SCR_017293) at a nominal magnification of SA29000 in super-resolution counting mode and yielding a physical pixel size of 0.82 Å (0.41 Å super-resolution pixel size) using a K3 Summit direct electron detector camera (Gatan). The total dose on the camera was set to be 60 e⁻ Å$^{-2}$ on the specimen, and each image was fractioned into 32 subframes, which were recorded with a defocus value at a range of −1.2 µm to −1.8 µm. The nominal magnification for the images of *Hs*TMEM120A-nanodisc sample was set to be SA22500 in super-resolution counting mode and yielding a physical pixel size of 1.07 Å (0.535 Å super-resolution pixel size). All the images were recorded by using a high-throughput beam-image shift data collection method (*Wu et al., 2019*). For cryo-EM data collection, the sample size represents the number of different grids loaded with the same batch of protein samples. Two different grids were used for collecting the cryo-EM data of *Hs*TMEM120A-nanodisc sample, whereas three different grids were used for collecting the data of *Hs*TMEM120A protein in the detergent.

## Image processing

Two datasets (datasets 1 and 2) were collected for the *Hs*TMEM120A-nanodisc sample. For dataset 1, a total of 4169 cryo-EM movies were aligned with dose-weighting using MotionCor2 program

(*Zheng et al., 2017*) (RRID:SCR_016499). Micrograph contrast transfer function (CTF) estimations were performed by using CTFFIND 4.1.10 program (*Rohou and Grigorieff, 2015*) (RRID:SCR_016732). The procedures described below were performed by using cryoSPARC v3.1 program (*Punjani et al., 2017*) (RRID:SCR_016501) unless stated otherwise. After manual inspection of the micrographs, 3940 were selected and 230 particles were picked manually from the micrograph and sorted into 2D classes. The best classes were selected and used as references for autopicking procedure. After the process, 3,837,005 particles were autopicked and extracted using a box size of 256 pixels. Six rounds of 2D classification were performed to remove ice spots, contaminants, aggregates, and obscure classes, yielding 269,838 particles for further refinement. The particles were classified into three classes using ab-initio reconstruction, and the reference models were generated by low-pass filtering of the maps of the three individual classes (including one well-resolved and two junk classes) to 20 Å for the subsequent refinement processes. Consequently, 64,716 particles corresponding to the well-resolved class were selected for further refinement. By using the reference model generated from the map of the prior well-resolved class, non-uniform (NU) refinement was performed with $C2$ symmetry imposed and resulted in a 6.2 Å cryo-EM density map.

In order to improve map resolution, a sequential heterogeneous refinement approach was applied according to the protocol described previously (*Zhang et al., 2017*). In detail, after the NU refinement, heterogeneous refinement with the unbiased reference models of the well-resolved and two junk classes (generated through the previously described ab-initio reconstruction step) was carried out to obtain a better density map. Next, the improved density map was subjected to low-pass processing at 15 Å and 30 Å to generate low-resolution models. They served as the references of resolution gradient for another heterogeneous refinement. Finally, the empty scaffold protein-lipid shell map was generated in Chimera (RRID:SCR_004097) by using the volume eraser tool. To obtain a noise-downscaled map, the whole protein-nanodisc complex map was subtracted with the region of peripheral scaffold protein-lipid shell downscaled with a scale factor of 0.5. By using the noise-downscaled map and empty scaffold-lipid shell as references for the next step of heterogeneous refinement, a 4.7 Å map was obtained from 29,085 particles. The above process was the first round of sequential heterogeneous refinement. To enlarge the dataset of good particles, the 269,838 particles produced by 2D classification were divided into nine subgroups, so that the number of particles in each subgroup is similar to the good particles yielding the 4.7 Å map. The good particles were combined with each individual subgroup, the duplicate particles were removed automatically, and every chimeric subgroup was further processed in a second round of sequential heterogeneous refinement. Subsequently, all good particles from each subgroup were combined (and the duplicate particles were removed) for a third round of sequential heterogeneous refinement, yielding a 4.2 Å density map from 93,224 particles. For dataset 2, a similar procedure was applied to yield 389,789 good particles, leading to reconstruction of a map at 3.8 Å. The two datasets of the *Hs*TMEM120A-nanodisc sample were merged and subjected to an additional round of 2D classification, the final round of sequential heterogeneous refinement, CTF refinement, and NU refinement with $C2$ symmetry applied. The combined data resulted in a 3.7 Å density map from 410,963 particles after the entire process.

For the *Hs*TMEM120A sample in the detergent, a total of 12,156 movie stacks were collected, followed by motion correction using MotionCor2 (*Zheng et al., 2017*) and CTF estimation using the graphics processing unit (GPU)-accelerated contrast transfer function (GCTF) program (*Zhang, 2016*) (RRID:SCR_016500). Particles were picked using Gautomatch (*Zhang, 2017*) and Topaz (*Bepler et al., 2019*) programs. Multiple rounds of 2D and 3D classifications were performed to clean up the particles. The first round of 3D classification generated eight classes, and class 7 (21.5%) showed well-resolved structural features including transmembrane helices and intracellular amphipathic helices. A second round of multi-reference 3D classification was then performed to improve the quality of the map, giving rise to a subset of 78.3% particles. The next round of 3D classification was conducted without image alignment, resulting in four classes. A 3D mask covering the protein region was applied to exclude the detergent micelle density. The class 1 (50.5%) and class 4 (14.7%) maps displayed continuous transmembrane and amphipathic helices and thus were selected for further processing. Particles were subsequently imported to cryoSPARC (*Punjani et al., 2017*) and the NU refinement was carried out to yield a map at 4.3 Å resolution. The final map was generated through a subsequent local refinement in cisTEM (*Grant et al., 2018*) (RRID:SCR_016502) and was reported at 4.0 Å resolution according to the gold standard Fourier shell correlation (GSFSC)

criterion. The local resolution of the map was estimated using the MonoRes program integrated in the Scipion framework (*Vilas et al., 2018*).

## Model building, refinement, and validation

Ab-initio model building of *Hs*TMEM120A was carried out in COOT 0.8.9 (*Emsley et al., 2010*) (RRID:SCR_014222) by referring to the 3.7 Å cryo-EM map and the secondary structure prediction by PSIPRED 4.0 (*Buchan and Jones, 2019*) (RRID:SCR_010246). The cryo-EM densities for the TMD were well defined and provided detailed features for model building and real-space refinement. Register of amino acid residues was guided mainly by the clearly defined densities of residues with bulky side chains such as Trp, Phe, Tyr, and Arg. The density of IL5 (residues 250–263) was insufficient for assignment of the side chains; only the backbone of the polypeptide chain was built for this region. For the N-terminal domain (residues 7–100), some local regions were not well resolved in the cryo-EM map, involving about 40% amino acid residues of the domain. While the backbones of these residues were built in the model by referring to the density feature at a high contour level (0.6034 V or 5.13 rmsd), their side chains remained uninterpreted.

For the structure of *Hs*TMEM120A in the detergent, the overall resolution of cryo-EM density map was lower than that of *Hs*TMEM120A-nanodisc complex. Nevertheless, the densities of TMD and HM were well-resolved, allowing assignment of ~60% amino acid side chains. The initial model of the TMD (residues 123–336) was built by referring to the corresponding domain of the *Hs*TMEM120A-nanodisc complex structure. Around 85 residues of the 213 residues in the TMD were tentatively built as alanine residues as there were no side-chain information in the map. As the density of the N-terminal domain showed almost only the polypeptide backbone without side chain information, the region (residues 7–100) was built as a poly-alanine model.

The models were refined against the corresponding cryo-EM maps by using the phenix.real_space_refine program (*Adams et al., 2010*) (RRID:SCR_014224) followed by manual adjustment in COOT to improve the overall geometry and fitting of models with the cryo-EM maps. The final models of the two *Hs*TMEM120A structures contained the bulk region of the protein covering amino acid residues 7–336. The pore profiles of the two structures were calculated by using HOLE program (*Smart et al., 1996*), and the narrowest site was defined as the location of the smallest pore radius. The electrostatic potential surface representations were calculated by ChimeraX (*Goddard et al., 2018*; *Pettersen et al., 2021*) (RRID:SCR_015872) or the APBS (*Baker et al., 2001*) plugin in PyMOL (The PyMOL Molecular Graphics System, Version 2.0 Schrödinger, LLC; RRID:SCR_000305). The statistics for data collection and processing, refinement, and validation of the two *Hs*TMEM120A structures are summarized in *Supplementary file 1*.

## Mass spectrometry

The endogenous ligand (CoASH) bound to *Hs*TMEM120A protein was extracted from the purified *Hs*TMEM120A protein sample by using perchloric acid (72%) as described in a previous study (*Shurubor et al., 2017*). The purified *Hs*TMEM120A protein (in detergent) with an $A_{260}/A_{280}$ value of 0.88 was concentrated to ~2 mg/ml. To extract the ligand, 1 ml *Hs*TMEM120A sample was mixed with perchloric acid to give a final concentration of perchloric acid of 5%. The mixture was incubated on ice for 10 min before being vortexed for 10–15 min and centrifuged at 21,000 ×g for 10 min at 4℃. The supernatant was removed and used for MS analysis immediately. The sample was analyzed by using a high-performance liquid chromatograph (HPLC) (Dionex Ultimate 3000) equipped with a high-resolution mass spectrometer (SCIEX TripleTOF 5600). The HPLC separation was carried out on an ACQUITY UPLC CSH C18 reversed-phase column (2.1 mm×100 mm, 1.7 μm [Waters]) at a flow rate of 0.25 ml/min. The column was maintained at 30℃. The mobile phase consisted of two components, namely phase A (50 mM ammonium acetate aqueous) and phase B (acetonitrile [ACN]). All solvents were of liquid chromatography–mass spectrometry (LC/MS) grade. The column was eluted for 0–3.1 min with a linear gradient from 0% to 5% B, 3.1–9.0 min from 5% to 25% B, 9.0–11.0 min from 25% to 40% B, and 11.0–15.0 min held at 40% B, followed by an equilibration step from 15.1 to 22 min at 0% B. The MS analysis was performed in positive ion mode at a resolution of 3000 for the full MS scan in an information-dependent acquisition (IDA) mode. The scan range for MS analysis was 50–1200 m/z with an accumulation time of 250 ms in the time-of-flight mass spectrometry (TOF MS) type and 100 ms in product ion type. Ion spray voltage was set at 5.5 kV in the positive mode,

and the source temperature was 600℃. The ion source gas 1 and 2 were both set at 60 psi, and the curtain gas was set at 35 psi. In the product ion mode, the collision energy was set at 35 V, while the collision energy spread was set at 15 V. Analysis of the LC/MS data was performed by using Peakview 2.1 software (RRID:SCR_015786). The identification of CoASH was achieved on the bases of three criteria, namely exact mass, product ion peak pattern, and isotope spectrum. Values for m/z were matched within five ppm to identify the CoASH. The exact mass was 768.12 based on the protonated molecular ion $[M^+H]^+$, $[C_{21}H_{36}N_7O_{16}P_3S^+H]^+$. The spectral data obtained from tandem mass spectrometry (MS/MS) showed two main product ion peaks at m/z 428.03 and m/z 261.13, corresponding to the fragments of CoASH.

## Isothermal titration calorimetry

The *Hs*TMEM120A protein was purified in the detergent conditions as described above. After gel filtration, the concentrated protein at 4.6 mg/ml was dialyzed overnight at 4℃ against 2 l dialysis buffer containing 50 mM NaCl, 20 mM HEPES (pH 7.5), 1 mM EDTA (pH 7.5), Protease Inhibitor Cocktail (MedChemExpress, 1 ml 100× stock solution per 150 ml solution), 0.006% GDN, 0.006% CHAPS, and 0.001% CHS. After concentration, the molar concentration of *Hs*TMEM120A monomer was 120 µM and $A_{260}/A_{280}$ was 0.58, and the molar concentration of W193A mutant monomer was 17.6 µM and $A_{260}/A_{280}$ was 0.62. Such a low $A_{260}/A_{280}$ value is close to the value (~0.6) for a pure protein, suggesting that the endogenous CoASH bound to *Hs*TMEM120A protein is largely removed during the purification process. For the ITC assay, the powder of coenzyme A (sodium salt hydrate; Sigma-Aldrich) was dissolved in the dialysis buffer to a final concentration of 6 mM as the stock solution and then subjected to serial dilution for the specific experimental requirements. All solutions were filtered with a 0.22-µm filter membrane and then degassed prior to use. Measurements of enthalpy change ($\Delta H^°$) upon CoASH binding were performed by a MicroCal ITC200 (Malvern). In order to ensure that the *Hs*TMEM120A protein was saturated with CoASH at a reasonable rate, the optimized ratio of protein to CoASH was adjusted to 1:1 (mol:mol). The sample cell was rinsed repeatedly by the dialysis buffer and 350 µl protein sample was loaded carefully into the cell to avoid formation of any bubbles. The injection syringe was filled with 120 µM CoASH solution for wild-type *Hs*TMEM120A or 17.6 µM CoASH solution for W193A mutant. The experimental temperature was set at 20℃. The total number of injections was 20, and the volume of each injection was 2 µl except that the first injection was of 0.4 µl. The CoASH solution was titrated into the *Hs*TMEM120A protein solution and the mixture was stirred at 700 rpm. The experimental data were analyzed by MicroCal ITC200 (RRID:SCR_020260). The heat change of the last four injections was so weak that the protein was regarded as completely saturated with CoASH despite that the heat produced did not go to zero. Thus, the heat generated by the last injection was subtracted from the heat generated by each injection. As a control, the dialysis buffer was titrated with CoASH under identical experimental conditions, and the weak background heat was also subtracted from the protein-CoASH binding data in the final results.

## Whole-cell electrophysiology and mechanical stimulation

The P1-KO-HEK293T cells (*Cahalan et al., 2015*) were grown in poly-D-lysine-coated coverslips in Dulbecco's modified Eagle medium (DMEM) supplemented with 10% fetal bovine serum (FBS), 100 units/ml penicillin, and 10 µg/ml streptomycin at 37℃ and 5% $CO_2$. DNA constructs including *Mm*Piezo1-mRuby2, *Mm*Piezo2-GST-ires-GFP, *Hs*TMEM63a-mCherry, *Mm*TMEM120A-mCherry, *Mm*TMEM120A-ires-GFP, *Hs*TMEM120A-mCherry, and vector were transfected using Lipofectamine 2000 (Thermo Fisher Technology). 24 hr or 36 hr after transfection and prior to electrophysiology, the cells were briefly digested with trypsin and sparsely re-plated onto poly-D-lysine-coated coverslips to obtain individual cells for recording. Electrophysiological recordings normally started about 2 hr later after re-plating the cells. The patch-clamp experiments were carried out with HEKA EPC10 as previously described (*Zhao et al., 2016*). For whole-cell patch-clamp recordings, the recording electrodes had a resistance of 2–6 MΩ when filled with the internal solution composed of (in mM) 133 CsCl, 1 $CaCl_2$, 1 $MgCl_2$, 5 ethylene glycol tetraacetic acid (EGTA), 10 HEPES (pH 7.3 with CsOH), 4 MgATP, and 0.4 $Na_2GTP$. The extracellular solution was composed of (in mM) 133 NaCl, 3 KCl, 2.5 $CaCl_2$, 1 $MgCl_2$, 10 HEPES (pH 7.3 with NaOH), and 10 glucose. All experiments were performed at room temperature. The currents were sampled at 20 kHz, filtered at 2 kHz using

the Patchmaster software (RRID:SCR_000034). Leak currents before mechanical stimulations were subtracted off-line from the current traces. Voltages were not corrected for the liquid junction potential.

Mechanical stimulation was delivered to the cell during the patch clamp being recorded at an angle of 80° using a fire-polished glass pipette (tip diameter 3–4 µm) as described. Downward movement of the probe toward the cell was driven by a Clampex-controlled piezo-electric crystal microstage (E625 LVPZT Controller/Amplifier; Physik Instrument). The probe had a velocity of 1 µm/ms during the downward and upward motion, and the stimulus was maintained for 150 ms. A series of mechanical steps in 1 µm increments was applied every 10 s, and currents were recorded at a holding potential of −60 mV.

## Cell-attached electrophysiology

Stretch-activated currents were recorded in the cell-attached or inside-out patch-clamp configuration using the HEKA EPC10 and the Patchmaster software as previously described (*Zhao et al., 2016*). The currents were sampled at 20 kHz and filtered at 1 kHz. The recording electrodes had a resistance of 2–5 MΩ when filled with a standard pipette solution consisting of (in mM) 130 NaCl, 5 KCl, 10 HEPES, 1 $CaCl_2$, 1 $MgCl_2$, and 10 tetraethylammonium chloride (TEA-Cl) (pH 7.3, balanced with NaOH). To maintain the membrane potential near 0 mV, the external solution consisted of (in mM) 140 KCl, 10 HEPES, 1 $MgCl_2$, and 10 glucose (pH 7.3 with KOH). All experiments were carried out at room temperature. Membrane patches with a seal resistance of at least 2 GΩ were held at −80 mV and stimulated with 200 ms or 500 ms negative pressure pulses given at either 5 or 10 mmHg steps from 0 to −120 mmHg through the recording electrode using a Patchmaster-controlled pressure clamp HSPC-1 device (ALA-scientific). Statistical analysis was performed in Prism 6 (RRID: SCR_002798) and statistical details can be found in *Figure 1* legend. Reported n values are labeled in *Figure 1B,D and F*. The sample sizes represent the number of cells tested in the whole-cell and cell-attached recordings as well as the inside-out patch configuration. The number of cells tested was determined based on the magnitude of the effect observed and the variance among data points. The sizes were chosen based on the consistency of data across different conditions and multiple experiments. Samples were grouped based on the plasmids transfected into the cells. The investigators were not blinded to group allocation.

## Single-channel electrophysiology on proteoliposome vesicles

The purified *Hs*TMEM120A protein was reconstituted into the membrane of GUVs by using a mixture of cholesterol:DPhPC:azolectin (w:w:w) at a ratio of 1:5:17 or 1:10:15 with a protein:azolectin ratio of 1:85 (w/w) through a modified sucrose method (*Battle et al., 2009*). The bath and pipette solutions contained 500 mM NaCl, 10 mM $CaCl_2$, and 10 mM HEPES-NaOH (pH 7.4). Patch pipettes with resistances of 5–6 MΩ were used, and the patch resistance was increased to ~2 GΩ after the pipette was sealed tightly with the GUV membrane. A negative pressure was applied through a Suction Control Pro pump (Nanion) with a stepwise or linear protocol while data were recorded at a constant holding potential. The data were acquired at 50 kHz with a 0.5-kHz filter and a HumBug 50/60 Hz Noise Eliminator (Quest Scientific), using an EPC-10 amplifier (HEKA). The Clampfit Version 10.0 (Axon Instruments) was used for data analysis and Igor Pro 6.37A (WaveMetrics; RRID:SCR_000325) was used for making the graphs. The sample size represents the number of patches tested in the inside-out recordings. The number of patches tested was determined based on the observation of channel activities upon application of the negative pressure. Samples were grouped based on the proteins of interest reconstituted into the GUVs. The investigators were not blinded to group allocation.

## Cell lines

The *Piezo1*-KO-HEK293 cells in which the endogenous *Piezo1* gene was disrupted were obtained from Dr. Ardem Patapoutian Laboratory (*Cahalan et al., 2015*). The cells were cultured in DMEM supplemented with 10% FBS, 100 U $ml^{-1}$ penicillin, and 100 µg $ml^{-1}$ streptomycin. *Piezo1*-KO-HEK293 cells were free from mycoplasma and other microorganisms.

## Acknowledgements

The cryo-EM data were collected at the Center for Biological Imaging (CBI), Core facilities for Protein Science at the Institute of Biophysics, Chinese Academy of Sciences. The sample screening time on Talos Arctica was sponsored by the National Laboratory of Biomacromolecules and CBI. We thank XJ Huang, BL Zhu, and XJ Li at CBI for their assistance in cryo-EM data collection; YY Chen, ZW Yang, and BX Zhou at IBP for their help with ITC experiments; NL Zhu and FQ Yang at the Laboratory of Proteomics, IBP, for the mass spectrometry analysis on the sample extracted from purified *Hs*TMEM120A protein; XB Liang for her support in sample preparation and data collection; YL Li from Peking University for sharing the *Hs*TMEM120A and *Hs*TMEM63A cDNAs; and Dr. E Gouaux's lab at the Vollum Institute, Oregon Health and Science University for the pEG BacMam vector. The project is funded by the National Natural Science Foundation of China (31925024 and 31670749 to ZL; 31825014 and 31630090 to BX) and the Strategic Priority Research Program of CAS (XDB37020101 to ZL and XDB37030304 to YZ).

## Additional information

### Funding

| Funder | Grant reference number | Author |
| --- | --- | --- |
| National Natural Science Foundation of China | 31925024 and 31670749 | Zhenfeng Liu |
| National Natural Science Foundation of China | 31825014 and 31630090 | Bailong Xiao |
| the Strategic Priority Research Program of CAS | XDB37020101 | Zhenfeng Liu |
| the Strategic Priority Research Program of CAS | XDB37030304 | Yan Zhao |

The funders had no role in study design, data collection and interpretation, or the decision to submit the work for publication.

### Author contributions

Yao Rong, Resources, Data curation, Formal analysis, Validation, Investigation, Visualization, Methodology, Writing - original draft, Writing - review and editing, Y.R. designed the expression construct of HsTMEM120A, expressed and purified the HsTMEM120A protein and reconstituted it in nanodiscs, extracted and identified CoASH through biochemical approaches, prepared the cryo-EM grids, collected cryo-EM data, built the model and refined the structure, carried out the ITC experiments. and also analyzed the structure; Jinghui Jiang, Resources, Formal analysis, Investigation, J.J. carried out electrophysiolocal recordings in cells and analyzed data; Yiwei Gao, Data curation, Formal analysis, Investigation, Visualization, Y.G. is involved in processing the cryo-EM data of HsTMEM120A-detergent sample; Jianli Guo, Data curation, Formal analysis, Investigation, Visualization, Writing - original draft, Writing - review and editing, J.G. reconstituted HsTMEM120A in GUV and performed single-channel recording experiments; Danfeng Song, Data curation, Formal analysis, Validation, Investigation, Visualization, Writing - review and editing, D.S. processed the cryo-EM data of HsTMEM120A-nanodisc sample; Wenhao Liu, Formal analysis, Investigation, W.L. carried out electrophysiological recordings in cells and analyzed data; Mingmin Zhang, Investigation, M.Z. cloned HsTMEM120A and HsTMEM63A and did initial electrophysiological characterizations of HsTMEM120A- and HsTMEM63A-mediated mechanically activated currents; Yan Zhao, Conceptualization, Data curation, Formal analysis, Supervision, Funding acquisition, Validation, Investigation, Writing - original draft, Writing - review and editing, Y.Z. conceived and coordinated the project, processed the cryo-EM data of HsTMEM120A-detergent sample, analyzed the structure; Bailong Xiao, Conceptualization, Resources, Data curation, Formal analysis, Supervision, Funding acquisition, Validation, Visualization, Writing - original draft, Writing - review and editing, B.X. conceived and coordinated the project; Zhenfeng Liu, Conceptualization, Data curation, Formal analysis, Supervision, Funding acquisition, Validation, Investigation, Visualization, Methodology, Writing - original

draft, Project administration, Writing - review and editing, Z.L. conceived and coordinated the project, and is involved in building and analyzing the structural model of HsTMEM120A and CoASH

## Author ORCIDs
Yao Rong  http://orcid.org/0000-0002-5590-2939
Yiwei Gao  http://orcid.org/0000-0001-8169-9332
Zhenfeng Liu  https://orcid.org/0000-0001-5502-9474

## Decision letter and Author response
Decision letter https://doi.org/10.7554/eLife.71474.sa1
Author response https://doi.org/10.7554/eLife.71474.sa2

## Additional files

### Supplementary files
• Supplementary file 1. Cryo-EM data collection and processing, refinement, and validation statistics of HsTMEM120A structures.

• Transparent reporting form

### Data availability
The structural models have been deposited in the Protein Data Bank under accession codes of 7F3T (https://www.rcsb.org/structure/7F3T) for HsTMEM120A-CoASH complex in nanodisc and 7F3U (https://www.rcsb.org/structure/7F3U) for HsTMEM120A in detergent. The cryo-EM density maps of HsTMEM120A-CoASH complex in nanodisc and HsTMEM120A in detergent have been deposited in the Electron Microscopy Data Bank under accession codes of EMD-31440 (https://www.emdataresource.org/EMD-31440) and EMD-31441 (https://www.emdataresource.org/EMD-31441), respectively. The source data files for Figure 1, Figure 1-figure supplement 1, Figure 2-figure supplement 1A, Figure 4D and Figure 4-figure supplement 4 have been provided.

The following datasets were generated:

| Author(s) | Year | Dataset title | Dataset URL | Database and Identifier |
|---|---|---|---|---|
| Song DF, Rong Y, Liu ZF | 2021 | Cryo-EM structure of human TMEM120A in the CoASH-bound state | https://www.rcsb.org/structure/7F3T | RCSB Protein Data Bank, 7F3T |
| Rong Y, Gao YW, Song DF, Zhao Y, Liu ZF | 2021 | Cryo-EM structure of human TMEM120A in the CoASH-free state | https://www.rcsb.org/structure/7F3U | RCSB Protein Data Bank, 7F3U |
| Song DF, Rong Y, Liu ZF | 2021 | Cryo-EM structure of human TMEM120A in the CoASH-bound state | https://www.emdatare-source.org/EMD-31440 | EMDataResource, EMD-31440 |
| Rong Y, Gao YW, Song DF, Zhao Y, Liu ZF | 2021 | Cryo-EM structure of human TMEM120A in the CoASH-free state | https://www.emdatare-source.org/EMD-31441 | EMDataResource, EMD-31441 |

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
