## [Decision Letter]

**Acceptance summary:**

The revised manuscript provides even stronger evidence that TMEM120A does not function as a mechanosensitive channel, together with cryo-EM data showing the structure of the protein in two different conformational states in the presence and absence of the metabolic co-factor CoASH. Together, the manuscript suggests a role for TMEM120A in lipid metabolism, opening interesting new paths for further research.

**Decision letter after peer review:**

Thank you for submitting your article "TMEM120A contains a specific coenzyme A-binding site and might not mediate poking-or stretch-induced channel activities in cells" for consideration by *eLife*. Your article has been reviewed by 3 peer reviewers, including Andrés Jara-Oseguera as the Reviewing Editor and Reviewer #1, and the evaluation has been overseen by Olga Boudker as the Senior Editor.

Essential revisions:

1) The authors should mention that the human TMEM120A construct that they used contains a covalently attached C-terminal mCherry tag that was not present in the original publication. Although unlikely, it remains a possibility that the tag prevents the protein from functioning as an ion channel when expressed in heterologous systems. Performing experiments with a construct without the tag would be desirable, but not required.

2) The methods state that hsTMEM120A was codon-optimized for insect cells (lines 432-434), but electrophysiological experiments were performed in mammalian cells. Were the constructs used in this experiment also codon-optimized for insect cells (rather than the construct used in the original TACAN paper)? This is unclear from the methods (lines 699-700). If so, please briefly discuss this as an alternative explanation for failure to replicate the mechanosensitive function with electrophysiology (i.e., due to poor translation and corresponding low membrane expression) in the discussion.

3) In Figure 1A-B, in addition to displaying the peak current for each cell, it would be useful to include a traditional displacement-current curve both for Piezos and for TMEMs- this would alleviate any concern that TMEM-transfected cells were not poked to the same depths. Similarly for Figure 1C, in addition to plotting maximal current please include P50 curves- because the authors smartly included a high-threshold channel as a positive control, it would further strengthen their claims to show that even at high pressures, there is no current above background for TMEM120A-B.

4) The authors should either tone down a majority of their detailed structural observations, or show the experimental density that supports them – this is particularly relevant for the CoASH binding site on Figure 4E, where residue-specific interactions are discussed; Figure 5C and D, where a conformational change is suggested without having shown the density for the loop on the CoASH-bound structure; the extracellular constriction formed by M207, W210, F219 and C310, for which also no density is shown. The depiction of isolated channel fragments as in Figure 3 – Supplement 1 is not helpful for addressing any of the issues described above.

5) The authors should edit their Introduction to make it more useful for wider audiences, mainly by clarifying that MscL and MscS are absent in animals, and by indicating the types of organisms that express the channels listed on lines 52-57.

6) A significant part of the Discussion section is dedicated to speculations on how TMEM120A might still function as a mechanosensitive channel in the absence of CoASH block, while the results of the study suggest that it might not even be a channel. It would be equally (or more) important to discuss its putative role in fatty acid metabolism, perhaps by referring to related/similar structures of the proteins with known functions (e.g. ELOVL). With respect to the functional role of TMEM120A, of interest might be recent preprints (https://doi.org/10.1101/2021.06.29.450322, https://doi.org/10.1101/2021.06.30.450616).

7) Line 450: please include the amount of membranes used for purification.

Line 455: please include the amount of Streptactin resin used.

Line 467: Please include typical yields.

Line 523: Please use a correct reference for SerialEM (Mastronarde, 2003).

Line 551: Please describe what served as a reference for NU refinement and how it was generated.

Line 554 and 556: Since ab initio reconstruction does not require a reference model, it would be unbiased.

Line 571: Please indicate at what point the duplicated particles were removed, and I would also include this in the processing scheme (Figure 2 —figure supplement 1).

Figure 5 —figure supplement 1 panel A – please include the number of particles at each stage of the processing workflow for consistency.

*Reviewer #3 (Recommendations for the authors):*

(1) "Despite that TACAN is crucial…" (lines 70-71) – the evidence provided on the Cell paper does not strongly establish a relevance for TMEM120A in nociception. We suggest the authors provide a more nuanced description of the observations made in the 2020 Cell paper, as this would be helpful for readers.

(2) Line 61: I suggest "exist in mammals".

Lines 191 – 193: I am not convinced that comparing TMEM120A structure to other mechanosensitive channels is necessary, as the protein is likely not a channel.

Line 316: shown

Line 318: "The drastic difference"

Line 598: Register

Figure 5. I would suggest rotating panels C and D by 180° so that they are in the same orientation as A. Alternatively, please indicate how C and D relate to A in the figure to better guide the reader.

(3) Figure 5 —figure supplement 2 panel B – it would be nice to see a pore radius of CoASH-free state for comparison.

---

## [Author Response]

Essential revisions:(1) The authors should mention that the human TMEM120A construct that they used contains a covalently attached C-terminal mCherry tag that was not present in the original publication. Although unlikely, it remains a possibility that the tag prevents the protein from functioning as an ion channel when expressed in heterologous systems. Performing experiments with a construct without the tag would be desirable, but not required.

We thank the reviewer for the thoughtful suggestion. In our study, we have tested both *Mm*TMEM120A-mCherry and *Mm*TMEM120A-ires-GFP, and observed neither poking- nor stretch-induced currents (Revised Figure 1C, F, G and I). For the *Mm*TMEM120A-mCherry construt, *Mm*TMEM120A is covalently fused to a C-terminal mCherry tag as the reviewer pointed out. On the other hand, for the *Mm*TMEM120A-ires-GFP construct, *Mm*TMEM120A and GFP were expressed as two independent proteins (not fused to each other) due to the presence of an internal ribosome entry site/ires. The results suggest that the lack of *Mm*TMEM120A-mediated currents is not dependent on the presence or absence of fusion tag. Moreover, we also tested a different construct with GFP fused to the N-terminal region of *Hs*TMEM120A (GFP-*Hs*TMEM120A) instead of C-terminal region. The result is similar to those with the *Hs*TMEM120A-mCherry constructs (Revised Figure 1D-F). Therefore, the lack of TMEM120A-mediated currents is not due to the influence of fusion tags. Besides, the mCherry and GFP proteins are fused to the C- or N-terminal region suspended in cytosol and they are distant from the transmembrane domain. We expected that the influence of the fusion tags on the protein function should be minimal. Accordingly, we reasoned that it is highly unlikely that the tag prevents the fusion protein from functioning as an ion channel if it did behave in the way as originally reported by Beaulieau-Laroche et al. We have included the points in the results and Discussion sections of the revised manuscript (lines 102-107, p5; lines 123-128, p6; lines 363-375, p22-23).

(2) The methods state that hsTMEM120A was codon-optimized for insect cells (lines 432-434), but electrophysiological experiments were performed in mammalian cells. Were the constructs used in this experiment also codon-optimized for insect cells (rather than the construct used in the original TACAN paper)? This is unclear from the methods (lines 699-700). If so, please briefly discuss this as an alternative explanation for failure to replicate the mechanosensitive function with electrophysiology (i.e., due to poor translation and corresponding low membrane expression) in the discussion.

We thank the reviewer for the great question and suggestion. The *Hs*TMEM120A-mCherry construct used in whole-cell and cell-attached electrophysiological experiments contains the cDNA amplified from the *Hs*TMEM120A clone in the cDNA library of human ORF 8.1 (a gift from Yulong Li's lab at the Peking University). It is different from the vector with the cDNA of *Hs*TMEM120A codon-optimized for protein expression in insect cell (for cryo-EM, single-channel electrophysiology and ITC experiments). Through gateway reactions, the coding sequence of *Hs*TMEM120A is cloned into the pDEST-mCherry expression vector, resulting in fusion of mCherry at the C-terminus of *Hs*TMEM120A. The cDNA of the *Mm*TMEM120A was amplified from a mouse cDNA library and subcloned into modified pEG BacMam vectors, yielding two constructs with the coding regions of mCherry-StrepII and ires-GFP fused to the 3’-region of the target gene product respectively (MmTMEM120A-mCherry and *Mm*TMEM120A-ires-GFP). Besides, we have also synthesized the cDNA of *Hs*TMEM120A (with codons optimized for expression in HEK293 cells) and cloned it into the pcDNA3.1 vector with an upstream GFP-encoding sequence (GFP-HsTMEM120A) for the whole-cell and cell-attached electrophysiology studies. We have added the detailed description of constructs and molecular cloning in the method section of the revised manuscript (lines 531-560, p35-36).

(3) In Figure 1A-B, in addition to displaying the peak current for each cell, it would be useful to include a traditional displacement-current curve both for Piezos and for TMEMs- this would alleviate any concern that TMEM-transfected cells were not poked to the same depths. Similarly for Figure 1C, in addition to plotting maximal current please include P50 curves- because the authors smartly included a high-threshold channel as a positive control, it would further strengthen their claims to show that even at high pressures, there is no current above background for TMEM120A-B.

We thank the reviewer for the constructive advices. In the revised Figure 1, we have included the displacement-current curves (revised Figure 1B) and pressure-current curves (revised Figure 1E and 1H).

(4) The authors should either tone down a majority of their detailed structural observations, or show the experimental density that supports them – this is particularly relevant for the CoASH binding site on Figure 4E, where residue-specific interactions are discussed; Figure 5C and D, where a conformational change is suggested without having shown the density for the loop on the CoASH-bound structure; the extracellular constriction formed by M207, W210, F219 and C310, for which also no density is shown. The depiction of isolated channel fragments as in Figure 3 – Supplement 1 is not helpful for addressing any of the issues described above.

Thank you for the suggestion. To show the cryo-EM densities corresponding to the structural parts mentioned by the reviewer, we have updated Figure 4 to include cryo-EM densities of the CoASH-binding site in panel E. The densities of IL5 loops in the *Hs*TMEM120A structures of CoASH-free and CoASH-bound states are included in Figure 5E, to provide evidences for the structural models shown in Figure 5C and D. Besides, the densities for the four amino acid residues (M207, W210, F219 and C310) forming the extracellular constrict are included in Figure 5−figure supplement 2C.

(5) The authors should edit their Introduction to make it more useful for wider audiences, mainly by clarifying that MscL and MscS are absent in animals, and by indicating the types of organisms that express the channels listed on lines 52-57.

In the revised manuscript, we have included a sentence to state that:

“While members of the MscL family are present in bacteria, archaea or fungi, and those of the MscS family exist in plants or microbes, they are not found in animals.” (lines 49-51, p3).

The types of organisms have also been indicated for the various mechanosensitive channels listed in the introduction (lines 54-59, p3).

(6) A significant part of the Discussion section is dedicated to speculations on how TMEM120A might still function as a mechanosensitive channel in the absence of CoASH block, while the results of the study suggest that it might not even be a channel. It would be equally (or more) important to discuss its putative role in fatty acid metabolism, perhaps by referring to related/similar structures of the proteins with known functions (e.g. ELOVL). With respect to the functional role of TMEM120A, of interest might be recent preprints (https://doi.org/10.1101/2021.06.29.450322, https://doi.org/10.1101/2021.06.30.450616).

Thanks a lot for your insightful suggestion. We have included a new paragraph to discuss the recent progresses on the putative functional role of TMEM120 in fatty acid metabolism and cited the related recent preprints (lines 392-406, p24).

(7) Line 450: please include the amount of membranes used for purification.Line 455: please include the amount of Streptactin resin used.Line 467: Please include typical yields.Line 523: Please use a correct reference for SerialEM (Mastronarde, 2003).Line 551: Please describe what served as a reference for NU refinement and how it was generated.Line 554 and 556: Since ab initio reconstruction does not require a reference model, it would be unbiased.Line 571: Please indicate at what point the duplicated particles were removed, and I would also include this in the processing scheme (Figure 2 —figure supplement 1).Figure 5 —figure supplement 1 panel A – please include the number of particles at each stage of the processing workflow for consistency.

In the revised manuscript, the amount of membranes used for purification, the amount of Streptactin resin used and typical yields have been included in the methods (line 587, p37; line 592, p38; lines 604-605, p38). The reference for Serial EM has been updated. The reference for NU refinement and how it was generated have been described in detail (lines 691-695, p42). The typo in “well-resolved and biased reference models” has been corrected as “the unbiased reference models of the well-resolved and two junk classes” (line 703, p42).

The point at which the duplicated particles were removed has been described in lines 718 (p43) and 721 (p43), and the labels of “duplicate particles removed” has been included in the revised Figure 2—figure supplement 1C.

In the updated version of in Figure 5—figure supplement 1, the number of particles at each stage of the processing workflow have been included in panel A.

Reviewer #3 Recommendations for the authors:(1) "Despite that TACAN is crucial…" (lines 70-71) – the evidence provided on the Cell paper does not strongly establish a relevance for TMEM120A in nociception. We suggest the authors provide a more nuanced description of the observations made in the 2020 Cell paper, as this would be helpful for readers.

Thank you very much for your suggestion. In the revised manuscript, we have rewritten the sentence as:

“Despite that the previous report suggested TACAN might function as a high-threshold mechanically activated cation channel responsible for sensing mechanical pain in mice, the mechanism of mechanosensation and ion permeation mediated by TACAN channel remains elusive and awaits to be investigated further.” (lines 75-80, p4).

(2) Line 61: I suggest "exist in mammals".Lines 191 – 193: I am not convinced that comparing TMEM120A structure to other mechanosensitive channels is necessary, as the protein is likely not a channel.Line 316: shownLine 318: "The drastic difference"Line 598: RegisterFigure 5. I would suggest rotating panels C and D by 180° so that they are in the same orientation as A. Alternatively, please indicate how C and D relate to A in the figure to better guide the reader.

In the revised manuscript, the typos have been corrected. Thanks a lot for your careful reading and pointing them out. We have also removed the description about comparing TMEM120A structure to other mechanosensitive channels and Figure 3−figure supplement 2 accordingly. Figure 5C and D have been updated to show the relationship between the view of panel C/D and that of panel A/B.

(3) Figure 5 —figure supplement 2 panel B – it would be nice to see a pore radius of CoASH-free state for comparison.

In the revised version of Figure 5—figure supplement 2, we have included the pore radius profile for the CoASH-free state for comparison with the one with CoASH bound. The profile of CoASH-bound structure looks different from the previous version because the Y axis is updated as the distance along the pore center line. The previous version contains an error in the values of Y axis (introduced during data analysis by accident) and has been fixed in the revised version.